# Robust Matrix Sensing in the Semi-Random Model

**Xing Gao**
**University of Illinois at Chicago**
xgao53@uic.edu

**Yu Cheng**
**Brown University**
yu_cheng@brown.edu

## Abstract

Low-rank matrix recovery is a fundamental problem in machine learning with numerous applications. In practice, the problem can be solved by convex optimization namely nuclear norm minimization, or by non-convex optimization as it is well-known that for low-rank matrix problems like matrix sensing and matrix completion, all local optima of the natural non-convex objectives are also globally optimal under certain ideal assumptions.

In this paper, we relax the assumptions and study new approaches for matrix sensing in a semi-random model where an adversary can add any number of arbitrary sensing matrices. More precisely, the problem is to recover a low-rank matrix $X^*$ from linear measurements $b_i = \langle A_i, X^* \rangle$, where an unknown subset of the sensing matrices satisfies the Restricted Isometry Property (RIP) and the rest of the $A_i$'s are chosen adversarially.

It is known that in the semi-random model, existing non-convex objectives can have bad local optima. To fix this, we present a descent-style algorithm that provably recovers the ground-truth matrix $X^*$. For the closely-related problem of semi-random matrix completion, prior work [CG18] showed that all bad local optima can be eliminated by reweighting the input data. However, the analogous approach for matrix sensing requires reweighting a set of matrices to satisfy RIP, which is a condition that is NP-hard to check. Instead, we build on the framework proposed in [KLL+23] for semi-random sparse linear regression, where the algorithm in each iteration reweights the input based on the current solution, and then takes a weighted gradient step that is guaranteed to work well locally. Our analysis crucially exploits the connection between sparsity in vector problems and low-rankness in matrix problems, which may have other applications in obtaining robust algorithms for sparse and low-rank problems.

## 1 Introduction

Low-rank matrix recovery is a popular inverse problem with many applications in machine learning such as collaborative filtering, image compression, and robust principal component analysis (PCA) [RS05, FCRP08, CLMW11]. In this paper, we study one of the most basic low-rank matrix recovery problems namely matrix sensing [CRT06, RFP10]. In the matrix sensing problem, we want to reconstruct a low-rank ground-truth matrix $X^* \in \mathbb{R}^{d_1 \times d_2}$ from a collection of sensing matrices $\{A_i\}_{i=1}^n$ and the corresponding linear measurements $b_i = \langle A_i, X \rangle$.

For notational convenience, we define a sensing operator $\mathcal{A}[\cdot] : \mathbb{R}^{d_1 \times d_2} \to \mathbb{R}^n$ such that $\mathcal{A}[X] = b$ with $b_i = \langle A_i, X \rangle$ for $i = 1 \ldots n$. The goal is to solve the following rank-constrained optimization problem:

$$\min_{X \in \mathbb{R}^{d_1 \times d_2}} \|\mathcal{A}[X] - b\|_2^2 \text{ subject to } \operatorname{rank}(X) \leq r \;.$$

As optimizing over low-rank matrices are often computationally hard, one common approach is to replace the non-convex low-rank constraint with its convex-relaxation, which results in the following

37th Conference on Neural Information Processing Systems (NeurIPS 2023).

nuclear norm minimization problem [RFP10]:

$$\min_{X \in \mathbb{R}^{d_1 \times d_2}} \|X\|_* \text{ subject to } \mathcal{A}[X] = b . \tag{1}$$

Another widely-used approach in practice is to consider the unconstrained non-convex factorized parametrization [RFP10, GJZ17, BNS16]:

$$\min_{U \in \mathbb{R}^{d_1 \times r}, V \in \mathbb{R}^{d_2 \times r}} \left\| \mathcal{A}[UV^\top] - b \right\|_2^2 . \tag{2}$$

and solve it with some form of gradient descent or alternating minimization.

Existing convex and non-convex approaches all rely on certain assumptions. A standard assumption in the literature is that the sensing matrices satisfy the Restricted Isometry Property (**RIP**), which means that the sensing matrices approximately preserve the norm of a low-rank matrix. (Formally, $\frac{1}{L} \cdot \|X\|_F^2 \leq \frac{1}{n} \sum_{i=1}^n \langle A_i, X \rangle^2 \leq L \cdot \|X\|_F^2$ given $\mathrm{rank}(X) \leq r$ for some parameters $r$ and $L$ .)

In this paper, we relax the RIP condition on the sensing matrices and study a robust version of the problem, which is often referred to as the **semi-random** model. More specifically, an adversary "corrupts" the input data by providing any number of additional sensing matrices $A_i$ that are adversarially chosen, but the corresponding measurements $b_i = \langle A_i, X^* \rangle$ remain consistent with the ground truth matrix $X^*$. Consequently, only a subset of the sensing matrices satisfy the RIP condition and the rest of them are arbitrary. This is an intermediate scenario between the average case and the worst case, which arises more frequently in practice.

To the best of our knowledge, we are the first to study the matrix sensing problem in this semi-random model. Formally, we consider the following adversary: suppose that originally there was a collection of RIP sensing matrices $\{A_i\}_{i=1}^m$ ("good" matrices), then the adversary augmented some arbitrary $\{A_i\}_{i=m+1}^n$ ("bad" matrices) and then shuffled all the sensing matrices. The algorithm is then given the measurements based on the "good" and "bad" matrices together. The combined sensing matrices are no longer guaranteed to satisfy the RIP condition, but there exists a subset (indicated by an indicator vector $w^*$) that does, i.e., $\frac{1}{L} \cdot \|X\|_F^2 \leq \sum_{i=1}^n w_i^* \langle A_i, X \rangle^2 \leq L \cdot \|X\|_F^2$, where $w_i^* = \frac{1}{m}$ on the original "good" matrices and $w_i^* = 0$ on the "bad" matrices added by the adversary. In general, the subset may be replaced by a convex combination and the indicator vector by a simplex. Inspired by the adversary for semi-random vector regression in [KLL+23], we refer to this condition as **wRIP** (weighted RIP) and formally define it in Definition 2.2.

Since the wRIP condition is a more general assumption than RIP, existing solutions that rely on RIP might fail under the semi-random model with wRIP condition. As stated in [KLL+23], this type of adversary does not break the problem from an "information-theoretic perspective", but affects the problem computationally. In particular, existing non-convex approaches for matrix sensing (e.g., 2) may get stuck at bad local minima as the RIP condition is necessary for proving landscape results regarding the non-convex objective (see, e.g., the counter-examples provided in [BNS16]. The convex relaxation approach (1) does continue to work in the semi-random model, because the augmented linear measurements are consistent with the ground-truth matrix $X^*$ which simply provides additional optimization constraints. However, convex approaches are often less desirable in practice and can become computationally prohibitive when $d_1, d_2 > 100$ as pointed out in [RFP10].

## 1.1 Our Contributions

The limitations of existing algorithms motivate us to pose and study the problem of semi-random matrix sensing in this paper. We summarize our main contributions below:

- **Pose and study matrix sensing in the semi-random model.** We introduce the more general wRIP condition on matrix sensing as a relaxation of the typical RIP assumption, and provide a solution that is more robust to input contamination. Our work will serve as a starting point for the design of more efficient robust algorithms for matrix sensing, as well as other low-rank matrix problems, in the semi-random model.

- **Design an efficient robust algorithm for semi-random matrix sensing.** Our algorithm is guaranteed to converge to a global optimum which improves on the existing non-convex solution [BNS16] that can get stuck in bad local optima in the semi-random model, while achieving a comparable running time as existing convex solution [RFP10], informally stated in Theorem 1.1 below. A formal statement can be found in Theorem 3.1.

- **Adapt a reweighting scheme for semi-random matrix sensing.** In contrast to the non-convex approach that failed and the convex approach that avoided the challenge posed by the adversary altogether, we study a new approach that directly targets the semi-random adversary instead. We develop an algorithm using an iterative reweighting approach inspired by [KLL+23]: in each iteration, the algorithm reweights the sensing matrices to combat the effect of the adversary and then takes a weighted gradient step that works well based on the current solution.

- **Exploit the connection between sparsity and low-rankness.** Observing a duality between sparse vectors and low-rank matrices, we draw a parallel between linear regression and matrix sensing problems. By exploring the structural similarities and differences between vector and matrix problems, we are able to extend and generalize the work of [KLL+23] on semi-random sparse vector recovery to the higher dimensional problem of semi-random matrix sensing. We emphasize that even though the generalization from vector to matrix problems is natural, the analysis behind the intuition is often nontrivial and involves different mathematical tools.

We state a simplified version of our main algorithmic result assuming Gaussian design. The more general result is stated as Theorem 3.1 in Section 3.

**Theorem 1.1** (Semi-Random Matrix Sensing). *Suppose the ground-truth matrix $X^* \in \mathbb{R}^{d_1 \times d_2}$ satisfies $\mathrm{rank}(X^*) \leq r$ and $\|X^*\|_F \leq \mathrm{poly}(d)$. Let $A_1, \ldots, A_n$ be the sensing matrices and let $b_i = \langle A_i, X^* \rangle$ be the corresponding measurements. Suppose there exists a (hidden) set of $\Omega(dr)$ sensing matrices with i.i.d. standard Gaussian entries, and the remaining sensing matrices are chosen adversarially, where $d = \max(d_1, d_2)$.*

*There exists an algorithm that can output $X \in \mathbb{R}^{d_1 \times d_2}$ such that $\|X - X^*\|_F \leq \epsilon$ with high probability in time $\widetilde{O}(nd^{\omega+1}r\log(1/\epsilon))$ [1] where $\omega < 2.373$ is the matrix multiplication exponent.*

## 1.2  Overview of Our Techniques

Since there exists a subset (or a convex combination in general) of the sensing matrices that satisfy the RIP condition, a natural strategy is to reverse the effect from the adversary by reweighting the sensing matrices so that they satisfy the RIP condition. However, it is NP-hard to verify RIP condition on all low-rank inputs, so it is unclear how to preprocess and "fix" the input in the beginning and then apply existing solutions to matrix sensing. Instead, we make a trade-off between the frequency of reweighting and the requirement on the weights by adopting an iterative reweighting approach: in each iteration, we only aim to find a set of weights so that the weighted matrices satisfy some desirable properties (not necessarily RIP) with respect to the current estimate $X$ (as opposed to all low-rank matrices).

Inspired by the workflow in [KLL+23], our **semi-random matrix sensing algorithm** (Algorithm 1) repeatedly calls a halving algorithm to reduce the error of our estimate arbitrarily small. The **halving algorithm** (Algorithm 2) contracts the upper bound on $\|X - X^*\|_F$, which is the error between our current estimate $X$ and the ground truth $X^*$, each time it is run. Inside this algorithm is a gradient descent style loop, where in each iteration we try to minimize a weighted objective function, which is essentially the weighted $\ell_2$-norm of $\mathcal{A}[X_t] - b$ (the distance to $X^*$ "measured" by the sensing matrices), where the weights are provided by an oracle implemented in Algorithm 3. The algorithm proceeds by taking a step opposite to the gradient direction, and the step is then projected onto a nuclear-norm-bounded ball which is necessary for the weight oracle to continue working in the next step. As we mentioned before, the weights from the oracle need to satisfy some nice properties with respect to the current iteration estimate $X_t$. Ideally, the property should: firstly, ensure the gradient step makes enough progress towards $X^*$; secondly, can be derived from the wRIP condition so that we know such a requirement is feasible; and lastly, be easily verifiable as opposed to the NP-hard RIP condition.

With the first requirement in mind, we define the **weight oracle** as in Definition 2.5. The oracle output should satisfy two properties, namely the progress and decomposition guarantees, and together they ensure the gradient step makes good enough progress toward $X^*$. Intuitively speaking, the progress guarantee ensures the gradient step is large in the direction parallel to the "actual" deviation $X - X^*$ (as opposed to only reducing the "measured" deviation $\mathcal{A}[X] - b$) and thus will make significant progress, while the decomposition guarantee ensures the gradient step has small contribution and

---

[1] Throughout the paper, we write $\widetilde{O}(f(n))$ for $O(f(n)\,\mathrm{polylog}\,f(n))$ and similarly for $\widetilde{\Omega}(\cdot)$.

effect in other directions thus will not cancel the progress after the projection. While the progress guarantee is quantified as an inner product, we introduce a concept called "norm-decomposition" (Definition 2.4) to capture the decomposition guarantee, and we will provide more details later. For the second requirement, we can loosely relate the two oracle guarantees to the wRIP condition: the (large) progress guarantee makes use of the lower bound in wRIP condition $\sum_{i=1}^{n} w_i^* \langle A_i, \frac{X}{\|X\|_F} \rangle^2 \geq \frac{1}{L}$, and the (small) decomposition guarantee makes use of the upper bound $\sum_{i=1}^{n} w_i^* \langle A_i, \frac{X}{\|X\|_F} \rangle^2 \leq L$.

We introduce a condition called **dRIP** (decomposable wRIP defined in Definition 2.3) to formally capture this relation, and we will show that it follows from the wRIP condition thus we can achieve such an oracle. Lastly, we will show that the oracle properties can be easily verified, meeting our third requirement.

A formal statement and a road map that leads to our main result can be found in Section 3.

## 1.3  Related Work

**Matrix sensing (RIP):** There are two main types of existing solutions. The convex-relaxation formulation 1 of the problem can be posed as a semidefinite program via the standard form primal-dual pair [RFP10], where the primal problem has a $(d_1 + d_2)^2$ semidefinite constraint and $n$ linear constraints. State of the art SDP solver [JKL+20] requires running time of $\widetilde{O}(nd^{2.5})$ where $d = \max(d_1, d_2)$. The other approach uses non-convex formulation 2 to reduce the size of the decision variable from $d^2$ to $dr$, improving computational efficiency. It is shown in [BNS16] that there are no spurious local minima given RIP sensing matrices and incoherent linear measurements in the non-convex approach, however, it is no longer applicable in the semi-random model.

**Semi-random model:** First introduced by [BS95], the semi-random model has been studied for various graph problems [FK01, PW17, MS10, MMV12]. Previously the work of [CG18] applied the semi-random model to the matrix completion problem, and recently [KLL+23] studied sparse vector recovery in this model.

**Semi-random matrix completion:** Low-rank matrix problems such as matrix completion and matrix sensing have similar optimization landscapes [GJZ17], thus development in one often lends insight to another. Prior work [CG18] on the closely-related problem of matrix completion under the semi-random model showed that all bad local optima can be eliminated by reweighting the input data via a preprocessing step. It exploits the connection between the observation data matrix and the Laplacian matrix of a complete bipartite graph, and gives a reweighting algorithm to preprocess the data in a black-box manner. However, the analogous approach for matrix sensing requires reweighting a set of matrices to satisfy RIP, which is a condition that is NP-hard to check, thus is not practical in the matrix sensing problem.

**Semi-random vector regression:** In order to overcome the barrier of the reweighting or preprocessing approach mentioned earlier, we take inspiration from the work of [KLL+23] on sparse vector recovery under the semi-random model. One of their main contributions is the "short-flat decomposition", which is a property that can be efficiently verified for a given vector (locally), instead of verifying the RIP condition for all sparse vectors (globally). They provide a projected gradient descent style algorithm, where the rows of the sensing matrix are reweighted differently in each iteration to ensure a "short-flat decomposition" exists for the gradient. We draw a parallel between the problem of sparse vector regression and low-rank matrix sensing, and extend their work on linear regression of sparse vectors to the more generalized problem of sensing low-rank matrices.

## 2  Preliminaries

### 2.1  Notations

Throughout the paper, we denote the ground-truth low-rank matrix as $X^*$. We assume $X^* \in \mathbb{R}^{d_1 \times d_2}$, $\text{rank}(X^*) = r$, and $d_1, d_2$ have the same order of magnitude. Let $d = \max(d_1, d_2)$.

We write $[n]$ for the set of integers $\{1, ..., n\}$. We use $\Delta^n$ for the nonnegative probability simplex in dimension $n$, and $\mathbb{R}_{\geq 0}^n$ for the set of vectors with nonnegative coordinates in $\mathbb{R}^n$. For a vector $x$, we denote its $\ell_1$, $\ell_2$, and $\ell_\infty$-norms as $\|x\|_1$, $\|x\|_2$ and $\|x\|_\infty$ respectively, and write the $i^{\text{th}}$ coordinate in $x$ as $x_i$. For a matrix $A$, we use $\|A\|_*$, $\|A\|_2$, and $\|A\|_F$ for the nuclear, spectral (operator), and

Frobenius norms of $A$ respectively. For a matrix $A$, we use $A_{(k)} = \mathrm{argmin}_{\mathrm{rank}(A') \leq k} \|A - A'\|_F$ to denote the best rank-$k$ approximation of $A$; or equivalently, given the SVD of $A = \sum_{i=1}^r \sigma_i u_i v_i^\top$, we have $A_{(k)} = \sum_{i=1}^k \sigma_i u_i v_i^\top$ where $\sigma_1, ..., \sigma_k$ are the top $k$ singular values of $A$.

We write $\mathrm{tr}(A)$ for the trace of a square matrix $A$. For matrices $A, B \in \mathbb{R}^{d_1 \times d_2}$, we write $\langle A, B \rangle$ for their entrywise inner product $\langle A, B \rangle = \langle A, B \rangle = \mathrm{tr}(A^\top B) = \sum_{j,k} A_{jk} B_{jk}$. A symmetric matrix $A \in \mathbb{R}^{d \times d}$ is positive semidefinite (PSD) if and only if $A = U^\top U$ for some matrix $U$, and we write $A \preccurlyeq B$ if $A$ and $B$ have the same dimension and $B - A$ is positive semidefinite. We write $\exp(A)$ as the matrix exponential of $A$; if $A$ is diagonalizable as $A = UDU^{-1}$ then $\exp(A) = U\exp(D)U^{-1}$.

## 2.2 Definitions

We formally define the matrix sensing operator and observation vector below.

**Definition 2.1** (Matrix Sensing Operator). Given a collection of sensing matrices $\mathcal{A} = \{A_i\}_{i=1}^n \subset \mathbb{R}^{d_1 \times d_2}$, we define the sensing operator $\mathcal{A}[\cdot] : \mathbb{R}^{d_1 \times d_2} \to \mathbb{R}^n$ such that $\mathcal{A}[X] = b$ where $b_i = \langle A_i, X \rangle$ for $X \in \mathbb{R}^{d_1 \times d_2}$.

In other words, we have $b := \sum_{i=1}^n \langle A_i, X \rangle e_i$ where $e_i$ is the $i^{\text{th}}$ standard basis vector in $\mathbb{R}^n$. Throughout the paper, we use either $\mathcal{A}$ or $\{A_i\}_{i=1}^n$ to represent the sensing matrices.

To consistently recover a rank-$r$ matrix in general, the number of measurements $n$ should be at least $(d_1 + d_2 - r)r$ [CP11], hence we assume $n = \widetilde{\Omega}(dr)$ where $\widetilde{\Omega}$ suppresses log factors. In most matrix sensing literature, it is standard to impose the Restricted Isometry Property (RIP) condition on the sensing matrices. The RIP condition states that $\mathcal{A}[\cdot]$ is approximately an isometry on low-rank matrices, which means the $\ell_2$-norm of the observation vector is close to the Frobenius norm of $X^*$.

In this paper, we consider a semi-random model and relax the RIP condition as follows: we require that there exist weights $\{w_i^*\}_{i=1}^n$ (or $w^* \in \Delta^n$) so that the weighted sensing matrices $\{\sqrt{w_i^*} A_i\}_{i=1}^n$ satisfy the RIP condition. We call this relaxed assumption the wRIP (weighted RIP) condition.

We formally define RIP and wRIP conditions below.

**Definition 2.2** (RIP and wRIP Conditions). We say a collection of sensing matrices $\mathcal{A} = \{A_i\}_{i=1}^n \subset \mathbb{R}^{d_1 \times d_2}$ satisfies the **RIP** (Restricted Isometry Property) condition with parameters $r$, $L$, and $\rho$ if the following conditions hold for all $X \in \mathbb{R}^{d_1 \times d_2}$ with $\mathrm{rank}(X) \leq r$:

1. Boundedness: $\|A_i\|_2 \leq \rho$ ;

2. Isometry: $\frac{1}{L} \cdot \|X\|_F^2 \leq \frac{1}{n} \sum_{i=1}^n \langle A_i, X \rangle^2 \leq L \cdot \|X\|_F^2$ .

Further, we say $\mathcal{A} = \{A_i\}_{i=1}^n$ satisfies the **wRIP** (weighted RIP) condition with parameters $r$, $L$, $\rho$, if $\exists w^* \in \Delta^n$ such that the following conditions hold for all $X \in \mathbb{R}^{d_1 \times d_2}$ with $\mathrm{rank}(X) \leq r$:

1. Boundedness: $\|A_i\|_2 \leq \rho$ ;

2. Isometry: $\frac{1}{L} \cdot \|X\|_F^2 \leq \sum_{i=1}^n w_i^* \langle A_i, X \rangle^2 \leq L \cdot \|X\|_F^2$ .

Notice that wRIP is a relaxation of the RIP condition, because we can choose $w_i^* = 1/n$ for all $i$ in the standard RIP setting. More importantly, wRIP is strictly weaker. For example, wRIP allows a (possibly majority) fraction of the sensing matrices to be chosen adversarially. We want to emphasize that the algorithm does not know $w^*$ — one of the main challenges of semi-random matrix sensing is that finding $w^*$ seems computationally hard, because it is NP-Hard to check the RIP condition.

For presenting our algorithm and analysis, we introduce a variant of the wRIP condition called dRIP (decomposable-wRIP).

**Definition 2.3** (dRIP Condition). We say a collection of sensing matrices $\mathcal{A} = \{A_i\}_{i=1}^n \subset \mathbb{R}^{d_1 \times d_2}$ satisfies the **dRIP** (decomposable wRIP) condition if $\exists w^* \in \Delta^n$ and constants $L, K, r, \rho \geq 1$, such that for all $V \in \mathbb{R}^{d_1 \times d_2}$ satisfying $\|V\|_F \in [\frac{1}{4}, 1]$, $\|V\|_* \leq 2\sqrt{2r}$:

1. Boundedness: $\|A_i\|_2 \leq \rho$ ;

2. Isometry: $\frac{1}{L} \leq \sum_{i=1}^n w_i^* \langle A_i, V \rangle^2 \leq L$ ;

3. Decomposability: $\exists(L, \frac{1}{K\sqrt{r}})$-norm-decomposition of $G^* = \sum_{i=1}^n w_i^* \langle A_i, V \rangle A_i = \sum_{i=1}^n w_i^* u_i A_i$ .

**Definition 2.4** (Norm Decomposition). We say a matrix $G$ has a $(C_F, C_2)$-norm-decomposition if $\exists S$ and $E$ s.t. $G = S + E$, and $\|S\|_F \leq C_F$, $\|E\|_2 \leq C_2$ .

The main difference with wRIP is that dRIP requires the additional "decomposition" property. Observe that $G^*$ is the (weighted) gradient at the point $V$. At a high level, we would like to decompose the gradient into two matrices, one with small Frobenius norm and the other one with small operator norm. Our matrix norm-decomposition is inspired by the "short-flat-decomposition" for vectors in [KLL$^+$23].

In Section 4, we will explain the motivation behind the norm decomposition as well as how to efficiently verify such a decomposition exists. We will also show that the dRIP condition is closely related to wRIP (by choosing parameters within a constant factor of each other) in Appendix C.

A crucial component in our algorithm is a weight oracle that produces a nonnegative weight on each sensing matrix (the weights are in general different in each iteration), such that the weighted gradient step moves the current solution closer to $X^*$. The oracle should output weights that satisfy certain properties which we term progress and decomposition guarantees. The purpose of these two guarantees is further explained in the proof of Lemma 4.2 in Appendix A.

**Definition 2.5** (Weight Oracle). We say an algorithm $\mathcal{O}$ is a $(C_{\text{prog}}, C_F)$-oracle, if given as input $n$ matrices $\mathcal{A} = \{A_i\}_{i=1}^n \subset \mathbb{R}^{d_1 \times d_2}$ and an vector $u = \mathcal{A}[V] \in \mathbb{R}^n$ where $V \in \mathbb{R}^{d_1 \times d_2}$, $\|V\|_F \in [\frac{1}{4}, 1]$, and $\|V\|_* \leq 2\sqrt{2r}$, the algorithm $\mathcal{O}(\mathcal{A}, u, \delta)$ returns a weight vector $w \in \mathbb{R}_{\geq 0}^n$ such that the following conditions hold with probability at least $1 - \delta$:

1. Progress guarantee: $\sum_{i=1}^n w_i u_i^2 \geq C_{\text{prog}}$ ;

2. Decomposition guarantee: $\exists(C_F, \frac{C_{\text{prog}}}{6\sqrt{r}})$ norm-decomposition of $G = \sum_{i=1}^n w_i u_i A_i$ .

Note that the progress guarantee is equivalent to $\langle G, V \rangle \geq C_{\text{prog}}$.

Finally we define numerical rank which we use in our analysis. Numerical rank serves as a lower bound for the rank of a matrix based on its nuclear norm and Frobenius norm. That is, we always have $\text{Rank}_n(A) \leq \text{rank}(A)$.

**Definition 2.6** (Numerical Rank). The numerical rank of $A$ is $\text{Rank}_n(A) = \frac{\|A\|_*^2}{\|A\|_F^2}$ .

## 3   Semi-Random Matrix Sensing

In this section, we present our main algorithm (Algorithm 1) for semi-random matrix sensing. Algorithm 1 with high probability recovers the ground-truth matrix $X^*$ to arbitrary accuracy.

---

**Algorithm 1:** SemiRandomMatrixSensing$(R_0, \epsilon, \delta, \mathcal{A}, b)$

---

1: Input: $R_0 \geq \|X^*\|_F, b = \mathcal{A}[X^*], \epsilon > 0, \delta \in (0, 1)$ ;
2: Output: $X_{\text{out}}$ s.t. $\|X_{\text{out}} - X^*\|_F \leq \epsilon$ .
3: $X_0 \leftarrow 0, T \leftarrow \log \frac{R_0}{\epsilon}, \delta' \leftarrow \frac{\delta}{T}, R \leftarrow R_0$ ;
4: **for** $0 \leq t \leq T$ **do**
5:     $X_{t+1} \leftarrow \text{HalveError}(X_t, R, \mathcal{O}, \delta', \mathcal{A}, b), R \leftarrow \frac{R}{2}$ ;
6: **end for**
7: Return $X_{\text{out}} \leftarrow X_T$ ;

---

The performance guarantee and runtime of Algorithm 1 are formally stated in the following theorem.

**Theorem 3.1** (Matrix Sensing under wRIP). *Suppose the ground-truth matrix $X^* \in \mathbb{R}^{d_1 \times d_2}$ satisfies* $\text{rank}(X^*) \leq r$ *and* $\|X^*\|_F \leq R_0$. *Suppose the sensing matrices $\mathcal{A} = (A_i \in \mathbb{R}^{d_1 \times d_2})_{i=1}^n$ satisfy* $(r, L, \rho)$-*wRIP (as in Definition 2.2). Let $b = \mathcal{A}[X^*] \in \mathbb{R}^n$ be the corresponding measurements.*

*For any $\epsilon, \delta > 0$, Algorithm 1 can output $X \in \mathbb{R}^{d_1 \times d_2}$ such that $\|X - X^*\|_F \leq \epsilon$ with probability at least $1 - \delta$. Algorithm 1 runs in time $O(nd^\omega \operatorname{polylog}(d) \log (\frac{L}{\delta} \log \frac{R_0}{\epsilon}) r \rho^2 L^4 \log \frac{R_0}{\epsilon})$ where $d = \max(d_1, d_2)$ and $\omega < 2.373$ is the matrix multiplication exponent.*

Theorem 1.1 is a direct corollary of Theorem 3.1 under Gaussian design.

*Proof of Theorem 1.1.* When there are $\Omega(dr)$ sensing matrices with i.i.d. standard Gaussian entries, the input sensing matrices satisfy $(r, L, \rho)$-wRIP for $L = O(1)$ and $\rho = O(d^{1/2})$ with probability at least $1 - \frac{1}{\operatorname{poly}(d)}$. This follows from a standard proof for RIP and the fact that we can ignore any sensing matrices with $\|A_i\|_2 \gg d^{1/2}$. We assume that the wRIP condition is satisfied.

By Theorem 3.1, when $L = O(1)$, $\rho = O(d^{1/2})$, $R_0 = \operatorname{poly}(d)$ and $\delta = \frac{1}{\operatorname{poly}(d)}$, Algorithm 1 can output a solution $X$ such that $\|X - X^*\|_F \leq \epsilon$ with high probability. The runtime of Algorithm 1 can be simplified to $\widetilde{O}(nd^{\omega+1} r \log(1/\epsilon))$. $\qquad\square$

We first provide a road map for our analysis for proving Theorem 3.1:

- Our main algorithm runs a "halving" subroutine for $\log \frac{R_0}{\epsilon}$ iterations to reduce the error to $\epsilon$. Each call to this subroutine reduces the upper bound on the distance between the current solution and the ground truth $X^*$ by half. This halving subroutine runs in time $O(nd^\omega \operatorname{polylog}(d) \log (\frac{L}{\delta} \cdot \log \frac{R_0}{\epsilon}) r \rho^2 L^4)$ according to Lemmas 4.2 and 4.3.

- In Section 4, we present the halving algorithm (Algorithm 2). It depends on a $(\Omega(1), O(1))$-oracle, and Lemma 4.1 shows that the oracle guarantees can be easily verified. The algorithm's correctness and running time are analyzed in Lemma 4.2 and Lemma 4.3.

- In Section 5 we present the weight oracle required by the halving algorithm. We first show in Lemma 5.1 that the wRIP condition implies that the sensing matrices satisfy the dRIP condition tailored to the design of the oracle. Then we present an implementation of the oracle in Algorithm 3 based on the dRIP condition, and analyze its correctness and running time in Lemma 5.3 and Lemma 5.4.

## 4 Algorithm for Halving the Error

In this section, we present Algorithm 2 (HalveError). Algorithm 2 takes an estimate $X_{\text{in}}$ with $\|X_{\text{in}} - X^*\|_F \leq R$ and outputs $X_{\text{out}}$ such that $\|X_{\text{out}} - X^*\|_F \leq \frac{R}{2}$. This is the matrix version of the HalfRaidusSparse [KLL+23] algorithm for vectors.

---

**Algorithm 2:** HalveError($X_{\text{in}}, R, \mathcal{O}, \delta, \mathcal{A}, b$)

1: Input: Rank-$r$ matrix $X_{\text{in}} \in \mathbb{R}^{d_1 \times d_2}$, $\|X_{\text{in}} - X^*\|_F \leq R$, $\mathcal{O}$ is a $(1, 12L^2)$-oracle for $\mathcal{A}$ with failure probability $\delta \in (0, 1)$, linear measurements $b = \mathcal{A}[X^*]$ .

2: Output: $X_{\text{out}} \in \mathbb{R}^{d_1 \times d_2}$ s.t. $\|X_{\text{out}} - X^*\|_F \leq \frac{R}{2}$ w.p. $\geq 1 - \delta$ and $\operatorname{rank}(X_{\text{out}}) \leq r$ .

3: $X_0 \leftarrow X_{\text{in}}$, $\mathcal{X} = \{X \in \mathbb{R}^{d_1 \times d_2} \mid \|X - X_{\text{in}}\|_* \leq \sqrt{2r}R\}$, $\eta = \frac{1}{288L^4}$, $T = \frac{6}{\eta}$ .

4: **for** $0 \leq t \leq T$ **do**

5: $\quad u_t \leftarrow \frac{1}{R}(\mathcal{A}[X_t] - b)$ ; $\qquad$ /* $u_t = \mathcal{A}[\frac{X_t - X^*}{R}]$ where $(u_t)_i = \frac{1}{R}\langle A_i, X_t - X^* \rangle$ */

6: $\quad w_t \leftarrow \mathcal{O}(\mathcal{A}, u_t, \frac{\delta}{T})$ ;

7: $\quad G_t \leftarrow \sum_{i=1}^n (w_t)_i (u_t)_i A_i$ ;

8: $\quad$ **if** $\mathcal{O}$ output satisfies the progress and decomposition guarantees on $u_t$ **then**

9: $\qquad X_{t+1} \leftarrow \operatorname{argmin}_{X \in \mathcal{X}} \|X - (X_t - \eta R G_t)\|_F^2$ ;

10: $\quad$ **else**

11: $\qquad$ Return $X_{\text{out}} \leftarrow (X_t)_{(r)}$ ; $\qquad$ /* Rank-r approximation of $X_t$ */

12: $\quad$ **end if**

13: **end for**

14: Return $X_{\text{out}} \leftarrow (X_T)_{(r)}$ ;

---

A crucial requirement of the algorithm is a $(\Omega(1), O(1))$-oracle for $\mathcal{A}$. In each iteration, the oracle takes a vector $u_t = \frac{\mathcal{A}[X_t]-b}{R}$, which is the (normalized) "measured deviation" between current estimate $X_t$ and $X^*$, and computes a weight vector $w_t$. The algorithm then tries to minimize the weighted objective function by gradient descent:

$$\text{Objective: } f_t(X) = \frac{1}{2}\sum_{i=1}^{n}(w_t)_i \langle A_i, \frac{X-X^*}{R}\rangle^2 \text{, i.e. } f_t(X_t) = \frac{1}{2}\sum_{i=1}^{n}(w_t)_i(u_t)_i^2 \text{,}$$

$$\text{Gradient: } \nabla_X f_t(X) = \sum_{i=1}^{n}(w_t)_i \langle A_i, \frac{X-X^*}{R}\rangle A_i \text{, i.e. } G_t = \nabla_X f_t(X)|_{X_t} = \sum_{i=1}^{n}(w_t)_i(u_t)_i A_i \text{.}$$

Ideally in the next iteration, we would like to make a step from $X_t$ in the opposite direction of the gradient $G_t$ with the goal of minimizing the deviation in the next iteration. However, we cannot take a step exactly in the direction of $G_t$, and our movement is constrained within a ball of (nuclear norm) radius $\sqrt{2r}R$ centered at $X_{\text{in}}$, namely the region $\mathcal{X} = \{X \mid \|X - X_{\text{in}}\|_* \le \sqrt{2r}R\}$. Nuclear norm is used as a proxy to control the rank and Frobenius norm of $X_t$ simultaneously throughout the algorithm: firstly, since $\|X_{\text{in}} - X^*\|_F \le R$, it makes sense that in each iteration $\|X_t - X_{\text{in}}\|_F \le R$ as well; secondly, while trying to minimize the difference between $X_t$ and $X^*$, we also want to ensure the rank of $X_t$ is relatively small, i.e. $\text{rank}(X_t) \le O(r)$. To tie things together, we use the following relationship between rank and numerical rank:

$$\text{rank}(X_t - X_{\text{in}}) \ge \text{Rank}_{\text{n}}(X_t - X_{\text{in}}) = \frac{\|X_t - X_{\text{in}}\|_*^2}{\|X_t - X_{\text{in}}\|_F^2} \text{.}$$

Assuming $\text{rank}(X_t) \ge \text{rank}(X_{\text{in}})$ and $\|X_t - X_{\text{in}}\|_F \le R$ throughout, then $\text{rank}(X_t) \ge \frac{\|X_t - X_{\text{in}}\|_*^2}{2R^2}$. Roughly speaking, in order for $\text{rank}(X_t) \le O(r)$, it is necessary that $\|X_t - X_{\text{in}}\|_* \le O(\sqrt{r}R)$, i.e. $X_t$ is inside some nuclear norm ball $\mathcal{X}$ of radius $O(\sqrt{r}R)$ centered at $X_{\text{in}}$. We set the radius of $\mathcal{X}$ to be $\sqrt{2r}R$ so that $X^* \in \mathcal{X}$, since $\|X_{\text{in}} - X^*\|_F \le R$, $\text{rank}(X_{\text{in}} - X^*) \le 2r$ therefore $\|X^* - X_{\text{in}}\|_* \le \sqrt{2r}R$. Thus we confine our movement within this nuclear norm ball of radius $\sqrt{2r}R$ centered at $X_{\text{in}}$ throughout the algorithm, and take the rank-$r$ approximation of the last $X_t$ to ensure $\text{rank}(X_{\text{out}}) \le r$ upon the termination of the algorithm.

To analyze the algorithm, first we show how to check whether the weight oracle output satisfies the progress and decomposition guarantees. The progress condition $\sum_{i=1}^{n} w_i u_i^2 \ge 1$ is trivial to verify, and we check whether $G$ is $(C_F, C_2)$-decomposable using Lemma 4.1, with details and proof deferred to Appendix A.

**Lemma 4.1** (Verify Norm Decomposition). *Given a matrix $G = U\Sigma V^\top = \sum_{i=1}^{d}\sigma_i u_i v_i^\top$ and $C_2 > 0$, suppose $\sigma_1 \ge ... \ge \sigma_k > C_2 \ge \sigma_{k+1}... \ge \sigma_d$, then for all $\|E\|_2 \le C_2$, we have $\|G - E\|_F^2 \ge \sum_{i=1}^{k}(\sigma_i - C_2)^2$.*

The following lemmas analyze the algorithm's correctness and show that it terminates with the desired distance contraction, as well as its running time. The proof is deferred to Appendix A.

**Lemma 4.2** (Algorithm 2: HalveError). *Given a $(1, 12L^2)$-oracle for $\mathcal{A}$ with failure probability $\delta \in (0, 1)$, where $\mathcal{A}$ satisfies the dRIP Condition 2.3, and $b = \mathcal{A}[X^*]$, Algorithm 2 succeeds with probability at least $1 - \delta$.*

**Lemma 4.3** (Algorithm 2 Running Time). *Algorithm 2 with failure probability $\delta$ runs in time $O(nd^\omega \text{polylog}(d) \log \frac{L}{\delta} r\rho^2 L^4)$.*

The crucial part of Lemma 4.2 shows that if current estimate $X_t$ is sufficiently far from $X^*$, i.e. $\|X_t - X^*\|_F \ge \frac{1}{4}R$, then according to Lemma 5.3 with high probability the weight oracle produces an output satisfying the progress and decomposition guarantees, and each iteration of Algorithm 2 decreases the distance to $X^*$ by a constant factor: $\|X_{t+1} - X^*\|_F^2 \le (1 - \frac{\eta}{2}) \cdot \|X_t - X^*\|_F^2$, thus after sufficient number of iterations the distance to $X^*$ will be halved. On the other hand, if the weight oracle fails, with high probability the current estimate $X_t$ is already sufficiently close to $X^*$, thus the algorithm can terminate early.

# 5   Oracle for Reweighting the Input

In this section, we present an algorithm (Algorithm 3) that serves as the weight oracle required by the error-halving algorithm (Algorithm 2). Algorithm 3 is the matrix version of the StepOracle [KLL+23] algorithm for vectors. We first state that, given proper choices of parameters within a constant factor, the wRIP Condition 2.2 implies the dRIP Condition 2.3, which is a more suitable property for our oracle implementation. The proof is deferred to Appendix C.

**Lemma 5.1** ($wRIP \implies dRIP$). *If $\mathcal{A}$ satisfies wRIP Condition 2.2 with parameters $r', L', \rho$, then $\mathcal{A}$ satisfies the dRIP Condition 2.3 with parameters $L, K, r, \rho$ such that $L = \Theta(L')$, $r = \Theta(r')$, and some constant $K \geq 1$.*

Now we are ready to present an implementation of the weight oracle in Algorithm 3 based on the dRIP condition. This algorithm takes as inputs the dRIP sensing matrices $\mathcal{A}$ and a vector $u$. If $u$ is an applicable input to the oracle, with high probability the algorithm outputs a weight vector $w$ satisfying the progress and decomposition guarantees as in Definition 2.5.

First we introduce some potential functions used in the algorithm.

**Definition 5.2** (Potential Functions in Algorithm 3). For sensing matrices $\mathcal{A} = \{A_i\}_{i=1}^n$ and input $u \in \mathbb{R}^n$ to the oracle, we define the following potential functions on weight vector $w \in \mathbb{R}^n$:

- Progress potential: $\Phi_{\text{prog}}(w) = \sum_{i=1}^n w_i u_i^2$ .

- Decomposition potential: $\Phi_{\text{dc}}(w) = \min_{\|S\|_F \leq L\|w\|_1} \left( \mu^2 \log \left[ F(G_w - S) \right] \right) + \frac{\|w\|_1}{4CLr}$ , where $G_w = \sum_{i=1}^n w_i u_i A_i$ and $F(E) = \text{tr} \exp \left( \frac{E^\top E}{\mu^2} \right)$ .

- Overall potential: $\Phi(w) = \Phi_{\text{prog}}(w) - Cr\Phi_{\text{dc}}(w)$ .

Note that $F(E) = \sum_{j=1}^d \exp \left( \frac{\sigma_j^2(E)}{\mu^2} \right)$ where $\sigma_j(E)$ is the $j^{\text{th}}$ singular value of $E$, due to properties of the exponential of a diagonalizable matrix.

The progress and decomposition potential functions control the progress and decomposition guarantees respectively, and later we will show that the termination condition is implied by the overall potential $\Phi \geq 0$. Consequently, by maximizing the overall potential each round, the algorithm tries to make as much progress as possible while ensuring $G$ is decomposable.

---

**Algorithm 3:** $\mathcal{O}(\mathcal{A}, u, \delta)$

---
1: Input: Sensing operator $\mathcal{A}$ satisfying dRIP Condition 2.3, $u \in \mathbb{R}^n$ .
2: Output: $w \in \mathbb{R}^n$ such that the algorithm is a $(1, 12L^2)$-oracle as in Definition 2.5 with probability $\geq (1 - \delta)$.
3: $C \leftarrow 108, \mu \leftarrow \frac{1}{\sqrt{Cr \log d}}, \eta \leftarrow \frac{1}{Kr\rho^2 \log d}, N' \leftarrow \log_2 \frac{1}{\delta}, N \leftarrow \frac{8Ln}{\eta}$ .
4: **for** $0 \leq k \leq N'$ **do**
5:    $w_0 \leftarrow 0$;
6:    **for** $0 \leq t \leq N$ **do**
7:       **if** $\Phi_{\text{prog}}(w_t) \geq 1$ **then**
8:          Return $w \leftarrow w_t$;
9:       **else**
10:          Sample $i \in [n]$ uniformly random;
11:          $s_t \leftarrow \text{argmax}_{s \in [0, \eta]} \Phi(w_t + se_i)$ ;
12:          $w_{t+1} \leftarrow w_t + s_t e_i$ ;
13:       **end if**
14:    **end for**
15: **end for**
16: Return $w \leftarrow 0$ ;

---

**Lemma 5.3** (Correctness of Algorithm 3). *Suppose $\mathcal{A}$ satisfies dRIP Condition 2.3 and $u$ is an applicable input to the weight oracle (that is, $u = \mathcal{A}[V] \in \mathbb{R}^n$ for some $V \in \mathbb{R}^{d_1 \times d_2}$ satisfying $\|V\|_F \in$*

$[\frac{1}{4}, 1]$ *and* $\|V\|_* \le 2\sqrt{2r}$*). Then, Algorithm 3 is a* $(1, 12L^2)$*-oracle for* $\mathcal{A}$ *(as in Definition 2.5) with failure probability at most* $\delta$.

We prove this lemma in two steps: first we show in Lemma B.1 that the output is valid; then in Lemma B.2 we show that the oracle achieves the success probability. Our weight oracle is inspired by the step oracle in [KLL+23]. It is worth noting that Lemma B.3, a key component used in the proof of Lemma B.2, is significantly different in the matrix case compared to the vector case. Lemma B.3 upper bounds the increase in $\Phi_{\mathrm{dc}}$ each round, which is then used to provide a lower bound for the increase in $\Phi$. Combining Lemma B.3 with our earlier remark that the algorithm terminates when $\Phi \ge 0$ gives us the number of iterations needed to terminate with high probability.

The running time of Algorithm 3 is stated in the following lemma, and the proof is deferred to Appendix B.

**Lemma 5.4** (Algorithm 3 Running Time)**.** *Algorithm 3 with failure probability* $\delta$ *runs in time* $O(nd^\omega \operatorname{polylog}(d) \log \frac{1}{\delta} r\rho^2)$.

# 6    Conclusion and Future Work

In this paper, we pose and study the matrix sensing problem in a natural semi-random model. We relax the standard RIP assumption on the input sensing matrices to a much weaker condition where an unknown subset of the sensing matrices satisfies RIP while the rest are arbitrary.

For this semi-random matrix sensing problem, existing non-convex objectives can have bad local optima. In this work, we employ an iterative reweighting approach using a weight oracle to overcome the influence of the semi-random input. Our solution is inspired by previous work on semi-random sparse vector recovery, where we exploit the structural similarities between linear regression on sparse vectors and matrix sensing on low-rank matrices.

Looking forward, we believe our approach can serve as a starting point for designing more efficient and robust algorithms for matrix sensing, as well as for other low-rank matrix and sparse vector problems in the semi-random model.

## Acknowledgement

We thank Rong Ge for helpful discussions. Xing Gao is supported in part by NSF awards ECCS-2217023 and CCF-2307106. Yu Cheng is supported in part by NSF Award CCF-2307106.

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

# A   Omitted Proofs from Section 4

To check whether $G$ is $(C_F, C_2)$-decomposable using Lemma 4.1, we first consider the following construction. Suppose the SVD of $G$ is $G = U\Sigma V^\top = \sum_{i=1}^d \sigma_i u_i v_i^\top$. Let $S = \sum_{i=1}^d \mu_i u_i v_i^\top$ with $\mu_i = \max(\sigma_i - C_2, 0)$ and let $E = \sum_{i=1}^d \lambda_i u_i v_i^\top$ with $\lambda_i = \min(C_2, \sigma_i)$. In other words, suppose $\sigma_1 \geq ... \geq \sigma_k > C_2 \geq \sigma_{k+1}... \geq \sigma_d$. For $i \leq k$ (i.e., $\sigma_i > C_2$), let $\mu_i = \sigma_i - C_2$ and $\lambda_i = C_2$; for $i > k$ (i.e., $\sigma_i \leq C_2$), let $\mu_i = 0$ and $\lambda_i = \sigma_i$. We have $G = S + E$ with $\|E\|_2 \leq C_2$, and $\|S\|_F = \sqrt{\sum_{i=1}^k (\sigma_i - C_2)^2}$. Then by Lemma 4.1, $G$ is $(C_F, C_2)$-norm-decomposable if and only if $\|S\|_F \leq C_F$ because:

- if $\|S\|_F \leq C_F$, we have a valid $(C_F, C_2)$-norm-decomposition for $G$ ;
- if $\|S\|_F > C_F$, a valid $(C_F, C_2)$-norm-decomposition does not exist for $G$ .

**Lemma 4.1** (Verify Norm Decomposition). *Given a matrix $G = U\Sigma V^\top = \sum_{i=1}^d \sigma_i u_i v_i^\top$ and $C_2 > 0$, suppose $\sigma_1 \geq ... \geq \sigma_k > C_2 \geq \sigma_{k+1}... \geq \sigma_d$, then for all $\|E\|_2 \leq C_2$, we have $\|G - E\|_F^2 \geq \sum_{i=1}^k (\sigma_i - C_2)^2$.*

*Proof.* Fix any $E$ with $\|E\|_2 \leq C_2$. Observe that for all $1 \leq i \leq k$, we have $u_i^\top G v_i = \sigma_i$ and $-C_2 \leq u_i^\top E v_i \leq C_2$. Consequently, for all $i \leq k$, we have $\sigma_i > C_2$ and $u_i^\top (G - E) v_i \geq \sigma_i - C_2 > 0$.

Let $S = G - E$, we have

$$\|S\|_F^2 = \|U^\top S V\|_F^2 = \sum_{i,j} (U^\top S V)_{ij}^2$$

$$\geq \sum_{i=1}^d (U^\top S V)_{ii}^2 \geq \sum_{i=1}^k (U^\top S V)_{ii}^2$$

$$= \sum_{i=1}^k (u_i^\top S v_i)^2 \geq \sum_{i=1}^k (\sigma_i - C_2)^2 . \qquad \square$$

**Lemma 4.2** (Algorithm 2: HalveError). *Given a $(1, 12L^2)$-oracle for $\mathcal{A}$ with failure probability $\delta \in (0, 1)$, where $\mathcal{A}$ satisfies the dRIP Condition 2.3, and $b = \mathcal{A}[X^*]$, Algorithm 2 succeeds with probability at least $1 - \delta$.*

*Proof.* We will first show that the distance to $X^*$ decreases by a constant factor after each iteration:

$$\|X_{t+1} - X^*\|_F^2 \leq \left(1 - \frac{\eta}{2}\right) \cdot \|X_t - X^*\|_F^2 .$$

Consider iteration $t$ in Algorithm 2: $X_{t+1} = \operatorname{argmin}_{X \in \mathcal{X}} \|X - (X_t - \eta R G_t)\|_F^2$. Taking the gradient of $\|X - (X_t - \eta R G_t)\|_F^2$ at $X = X_{t+1}$, we get $2[X_{t+1} - (X_t - \eta R G_t)]$. Since $X^* \in \mathcal{X}$ and $X_{t+1}$ is the local minimizer in $\mathcal{X}$:

$$2\langle X_{t+1} - (X_t - \eta R G_t), X_{t+1} - X^* \rangle \leq 0 .$$

Rearranging the terms gives

$$2\eta R \langle G_t, X_{t+1} - X^* \rangle \leq -2 \langle X_{t+1} - X_t, X_{t+1} - X^* \rangle .$$

To simplify, let $D = X_{t+1} - X_t$, $D_t = X_t - X^*$, $D_{t+1} = X_{t+1} - X^*$. Note that $D + D_t = D_{t+1}$ .

$$2\eta R \langle G_t, D_{t+1} \rangle \leq -2 \langle D, D_{t+1} \rangle = \langle D - D_{t+1}, D - D_{t+1} \rangle - \langle D, D \rangle - \langle D_{t+1}, D_{t+1} \rangle$$
$$= \langle D_t, D_t \rangle - \langle D, D \rangle - \langle D_{t+1}, D_{t+1} \rangle ,$$
$$\|D_t\|_F^2 - \|D_{t+1}\|_F^2 \geq 2\eta R \langle G_t, D_{t+1} \rangle + \langle D, D \rangle = 2\eta R \langle G_t, D_t \rangle + 2\eta R \langle G_t, D \rangle + \langle D, D \rangle . \quad (3)$$

Inequality (3) provides a lower bound on the distance contraction after each iteration. We break the right-hand side into two parts. The first term $2\eta R \langle G_t, D_t \rangle$ corresponds to the magnitude of the step $G_t$ in the direction $D_t = X_t - X^*$, which is the progress made by this step. To lower bound it, we

will use the **progress guarantee** of the $(1, 12L^2)$-oracle. Recall $u_t = \frac{1}{R}(\mathcal{A}[X_t] - b)$ and consider $V_t = \frac{1}{R}(X_t - X^*) = \frac{1}{R}D_t$ so that $u_t = \mathcal{A}[V_t]$. Given that the oracle's output satisfies the progress guarantee, which states that $\sum_{i=1}^{n}(w_t)_i(u_t)_i^2 \geq 1$, we have:

$$2\eta R \langle G_t, D_t \rangle = 2\eta R^2 \langle G_t, V_t \rangle = 2\eta R^2 \Big\langle \sum_{i=1}^{n}(w_t)_i(u_t)_i A_i, V_t \Big\rangle = 2\eta R^2 \sum_{i=1}^{n}(w_t)_i(u_t)_i^2 \geq 2\eta R^2 \ . \tag{4}$$

The remaining term $2\eta R \langle G_t, D \rangle + \langle D, D \rangle$ might be negative and cancel some of our progress. A natural attempt is to try to bound it using $2\eta R \langle G_t, D \rangle + \langle D, D \rangle \geq -\eta^2 R^2 \langle G_t, G_t \rangle$. However, the wRIP condition of $\mathcal{A}$ does not provide any guarantee on $\|G_t\|_F^2$. (In fact, we can derive that $\|G\|_2 \leq L$ from wRIP, but the best we can hope for is $\|G\|_F^2 \leq \text{rank}(G) \cdot \|G\|_2^2$ where $\text{rank}(G) \leq d$.) This motivates the decomposition property in the dRIP Condition 2.3 and in the weight oracle. The idea is that even though we cannot directly bound $\|G_t\|_F$, we can in fact lower bound $2\eta R \langle G_t, D \rangle + \langle D, D \rangle$ the term by decomposing $G_t$ into a Frobenius-norm-bounded matrix $S_t$, and an operator-norm-bounded matrix $E_t$. Specifically, we will use the **decomposition guarantee** of the $(1, 12L^2)$-oracle, which states that there exists norm-decomposition of $G_t = S_t + E_t$ where $\|S_t\|_F \leq 12L^2$ and $\|E_t\|_2 \leq \frac{1}{6\sqrt{r}}$. As our movement is confined in $\mathcal{X}$, $D = X_{t+1} - X_t$ is nuclear-norm-bounded so the inner product $\langle E_t, D \rangle$ can be bounded by generalized Holder's inequality. Recall $\eta = \frac{1}{288L^4}$.

$$\begin{aligned}
2\eta R \langle G_t, D \rangle + \langle D, D \rangle &= 2\eta R \langle E_t, D \rangle + 2\eta R \langle S_t, D \rangle + \langle D, D \rangle \\
&\geq -2\eta R \|E_t\|_2 \cdot \|D\|_* - \eta^2 R^2 \langle S_t, S_t \rangle \\
&\geq -2\eta R \cdot \frac{1}{6\sqrt{r}} \cdot 2\sqrt{2r}R - \eta^2 R^2 \cdot 144L^4 \\
&\geq -\frac{3}{2}\eta R^2 \ . \tag{5}
\end{aligned}$$

Putting inequalities 3,4 and 5 together:

$$\|D_t\|_F^2 - \|D_{t+1}\|_F^2 \geq 2\eta R^2 - \frac{3}{2}\eta R^2 \geq \frac{\eta}{2} \cdot \|D_t\|_F^2 \implies \|D_{t+1}\|_F^2 \leq \Big(1 - \frac{\eta}{2}\Big) \cdot \|D_t\|_F^2 \ .$$

In the case that the algorithm terminates after $T = \frac{6}{\eta}$ iterations,

$$\|X_T - X^*\|_F^2 \leq \Big(1 - \frac{\eta}{2}\Big)^T \cdot \|X_{\text{in}} - X^*\|_F^2 \leq \exp\Big(-\frac{\eta T}{2}\Big) \cdot R^2 \leq \frac{1}{16}R^2 \ ,$$

$$\|X_{\text{out}} - X^*\|_F \leq \|X_{\text{out}} - X_T\|_F + \|X_T - X^*\|_F \leq 2\|X_T - X^*\|_F \leq \frac{1}{2}R \ .$$

The last inequality comes from $X_{\text{out}}$ being the best rank-$r$ approximation of $X_T$.

In the case that the algorithm terminates early at Line 11, we can assume with probability at least $1 - \frac{\delta}{T}$ that the weight oracle would have succeeded given applicable input $u_t$. Then failure to satisfy the progress and decomposition guarantees means that $u_t$ is not an applicable input, which means $V_t$ does not satisfy the norm constraint in the weight oracle. $\|V_t\|_* \leq 2\sqrt{2r}$ is guaranteed because $X_t \in \mathcal{X}$, and $\|V_t\|_F = \frac{1}{R}\|X_t - X^*\|_F$ is decreasing in each round, so we must have $\|V_t\|_F < \frac{1}{4}$, which means $\|X_t - X^*\|_F < \frac{1}{4}R$. By the same argument as above, $\|X_{\text{out}} - X^*\|_F \leq \frac{1}{2}R$.

Finally, by a union bound on the failure probability of the weight oracle, the algorithm succeeds with probability at least $1 - \delta$. $\qquad\square$

**Lemma 4.3** (Algorithm 2 Running Time). *Algorithm 2 with failure probability $\delta$ runs in time $O(nd^\omega \, \text{polylog}\,(d) \log \frac{L}{\delta} r \rho^2 L^4)$ .*

*Proof.* Algorithm 2 has a for-loop that's repeated for $T = O(L^4)$ times.

Inside the loop, line 5 and 7 takes linear time $O(nd^2)$. Computing $w_t$ using the oracle (line 6) runs in time $O(nd^\omega \, \text{polylog}\,(d) \log \frac{L}{\delta} r \rho^2)$ according to Lemma 5.4. Line 8 through line 12 are all upper

bounded by time of SVD, which is on the same order of matrix multiplication $O(d^\omega)$ [DDH07], with current best of $O(d^{2.373})$[Wil14]. In particular, verifying the oracle guarantees (line 8) can be solved as an eigenvalue problem. Finding $X_{t+1}$ (line 9) is equivalent to projecting $X'_{t+1} := \frac{X_t - \eta R G_t - X_{\text{in}}}{\sqrt{2rR}}$ onto the unit nuclear norm ball. We first perform SVD on $X'_{t+1}$ then binary search for the largest truncation from its singular values to reach the nuclear norm sphere in time $O(d \log d)$, and the entire projection step is dominated by SVD. Finally the output step consists of SVD and matrix multiplication.

The overall running time of the algorithm is dominated by the weight oracle, so the total running time is $O(nd^\omega \operatorname{polylog}(d) \log \frac{L}{\delta} r \rho^2 L^4)$. $\qquad\square$

# B  Omitted proofs from Section 5

First we state a couple of lower bounds related to the decomposition potential function.

**Claim B.1.** $\mu^2 \log[F(E)] \geq \mu^2 \log(d)$ *and* $\mu^2 \log[F(E)] \geq \|E\|_2^2$.

*Proof.* For the first lower bound, $\exp\left(\frac{\sigma_j^2(E)}{\mu^2}\right) \geq 1$ for all $j \in [d]$ so $F(E) = \sum_{j=1}^d \exp\left(\frac{\sigma_j^2(E)}{\mu^2}\right) \geq d$.

For the second lower bound, $F(E) = \sum_{j=1}^d \exp\left(\frac{\sigma_j^2(E)}{\mu^2}\right) \geq \exp\left(\frac{\sigma_1^2(E)}{\mu^2}\right) = \exp\left(\frac{\|E\|_2^2}{\mu^2}\right)$. $\qquad\square$

**Lemma B.1** (Correctness of Algorithm 3). *If Algorithm 3 terminates from the inner loop, the output satisfies the progress and decomposition guarantees as defined in 2.5.*

*Proof.* We start with $w_0 = 0$, which means $\Phi_{\text{prog}}(w_0) = 0$, $\Phi_{\text{dc}}(w_0) = \mu^2 \log d$, and $\Phi(w_0) = 0 - Cr\mu^2 \log d = -1$.

At each round, since $s_t$ is chosen to maximize $\Phi(w_t + s_t e_i)$, in particular if we choose $s_t = 0$ then $\Phi(w_{t+1}) = \Phi(w_t)$, so $\Phi(w_{t+1}) \geq \Phi(w_t)$ which is non-decreasing. By definition $\Phi_{\text{prog}}(w_t)$ is also non-decreasing, and increases by at most 1 each round, because $\Phi_{\text{prog}}(w_{t+1}) - \Phi_{\text{prog}}(w_t) = s_t u_i^2 \leq \eta(\|A_i\|_2 \|V\|_*)^2 \leq 8\eta r \rho^2 \leq \frac{8}{K \log d} \leq 1$. $\Phi_{\text{dc}}(w_t)$ may not be monotone, but we have $\Phi_{\text{dc}}(w_t) \geq \mu^2 \log d$.

Suppose the algorithm terminates at round $t$ during one of the inner loops, which means $\Phi_{\text{prog}}(w_{t-1}) < 1$ and $1 \leq \Phi_{\text{prog}}(w_t) < 2$.

**Progress guarantee**: $\Phi_{\text{prog}}(w_t) = \sum_{i=1}^n (w_t)_i u_i^2 \geq 1$ is satisfied upon termination.

**Decomposition guarantee**:

$$\Phi(w_t) = \Phi_{\text{prog}}(w_t) - Cr\Phi_{\text{dc}}(w_t) \geq \Phi(w_0) = -1,$$

$$\implies \min_{\|S\|_F \leq L\|w\|_1} \left( \mu^2 \log[F(G_w - S)] \right) + \frac{\|w_t\|_1}{4CLr} = \Phi_{\text{dc}}(w_t) \leq \frac{\Phi_{\text{prog}}(w_t) + 1}{Cr} \leq \frac{3}{Cr},$$

$$\min_{\|S\|_F \leq L\|w_t\|_1} \left( \mu^2 \log[F(G_{w_t} - S)] \right) \leq \frac{3}{Cr} \implies \exists \|S\|_F \leq L\|w_t\|_1 \; s.t. \; \|G_{w_t} - S\|_2^2 \leq \frac{3}{Cr},$$

$$\text{and } \frac{\|w_t\|_1}{4CLr} \leq \frac{3}{Cr} \implies \|w_t\|_1 \leq 12L.$$

So there exist $\|S\|_F \leq 12L^2$ and $\|E\|_2 = \|G - S\|_2 \leq \frac{\sqrt{3}}{\sqrt{Cr}} = \frac{1}{6\sqrt{r}}$ which satisfy the decomposition guarantee.

Notice that for any round $t' < t$, $\Phi_{\text{prog}}(w_{t'}) < 1$, we also have $\Phi_{\text{dc}}(w_{t'}) \leq \frac{\Phi_{\text{prog}}(w_{t'}) + 1}{Cr} \leq \frac{2}{Cr}$, so $\Phi_{\text{dc}}(w_t) \leq \frac{3}{Cr}$ throughout the algorithm, which is a fact we will use later in Lemma B.3. $\qquad\square$

**Lemma B.2** (Success probability of Algorithm 3). *Given $\mathcal{A}$ satisfying dRIP Condition and applicable input $u$, Algorithm 3 terminates from the inner loop with probability at least $1 - \delta$.*

*Proof.* We first show the probability that the algorithm terminates from the inner loop is at least $\frac{1}{2}$, i.e., $\Pr[\Phi_{\text{prog}}(w_t) \geq 1] \geq \frac{1}{2}$ for some $t \leq N$.

Notice that $\Phi(w_t) = \Phi_{\text{prog}}(w_t) - Cr\Phi_{\text{dc}}(w_t) \geq 0 \implies \Phi_{\text{prog}}(w_t) \geq Cr\Phi_{\text{dc}}(w_t) \geq Cr\Phi_{\text{dc}}(w_0) = 1$, therefore the algorithm starts with $\Phi(w_0) = -1$ and will terminate once $\Phi(w_t) \geq 0$. Also notice that throughout the algorithm $\Phi(w_t) < 1$ because $\Phi_{\text{prog}}(w_t) < 2$ and $Cr\Phi_{\text{dc}}(w_t) \geq 1$ (from proof of Lemma B.1).

To prove by contradiction, assume that $\Pr[\Phi_{\text{prog}}(w_t) \geq 1] < \frac{1}{2}$ for all $t \leq N$, i.e., $\Pr[\text{continue}] \geq \frac{1}{2}$ for all rounds. We will lower bound the expected increase in $\Phi(w_t)$ each round, and we will show that with sufficiently large $N$, $\mathbb{E}[\Phi(w_N)] \geq 1$ contradicting $\Phi(w_t) < 1$ for all $t \leq N$.

Recall that $\Phi(w_t) = \Phi_{\text{prog}}(w_t) - Cr\Phi_{\text{dc}}(w_t)$, the lower bound for increase in $\Phi_{\text{prog}}(w_t)$ is provided by dRIP condition on $\mathcal{A}$ and applicable input $u$. The upper bound for expected increase in $\Phi_{\text{dc}}(w_t)$ is provided by Lemma B.3.

Given the algorithm continues at round $t \leq N$, consider choosing $s_t = \eta w_i^*$ so that $w' = w_t + \eta w_i^* e_i$, then the expected increase in $\Phi$ is at least:

$$\begin{aligned}
\mathbb{E}[\Phi(w_{t+1}) - \Phi(w_t) \mid \text{continue}] &= \mathbb{E}[\Phi_{\text{prog}}(w_{t+1}) - \Phi_{\text{prog}}(w_t)] - Cr\mathbb{E}[\Phi_{\text{dc}}(w_{t+1}) - \Phi_{\text{dc}}(w_t)] \\
&\geq \mathbb{E}[\Phi_{\text{prog}}(w') - \Phi_{\text{prog}}(w_t)] - Cr\mathbb{E}[\Phi_{\text{dc}}(w') - \Phi_{\text{dc}}(w_t)] \\
&= \frac{1}{n}\sum_{i=1}^{n} \eta w_i^* u_i^2 - Cr\Big(\mathbb{E}[\Phi_{\text{dc}}(w')] - \Phi_{\text{dc}}(w_t)\Big) \\
&\geq \frac{\eta}{Ln} - Cr \cdot \frac{\eta}{2CLnr} \\
&= \frac{\eta}{2Ln} .
\end{aligned}$$

Given the algorithm stops after round $t$, $\mathbb{E}[\Phi(w_{t+1}) - \Phi(w_t) \mid \text{stop}] = 0$. Overall:

$$\begin{aligned}
\mathbb{E}[\Phi(w_{t+1}) - \Phi(w_t)] &= \mathbb{E}[\Phi(w_{t+1}) - \Phi(w_t) \mid \text{continue}] \cdot \Pr[\text{continue}] + 0 \\
&\geq \frac{\eta}{2Ln} \cdot \Pr[\text{continue}] .
\end{aligned}$$

By choosing a sufficiently large $N = \frac{8Ln}{\eta}$:

$$\mathbb{E}[\Phi(w_N)] \geq \Phi(w_0) + \frac{\eta N}{2Ln} \cdot \Pr[\text{continue}] \geq -1 + \frac{\eta N}{4Ln} \geq 1 ,$$

contradicting $\Phi(w_t) < 1$. This means each inner loop of the algorithm will terminate with probability greater than $\frac{1}{2}$. Finally, we boost the success probability to $(1-\delta)$ using the outer loop with $N' = \log_2 \frac{1}{\delta}$ iterations. $\qquad\square$

Lemma B.3 provides a crucial bound used in the proof. Even though it achieves similar result as Lemma 13 [KLL+23] on the potential functions defined for vectors, analyzing the potential function defined for matrices involves very different techniques.

**Lemma B.3** (Potential Increase Upper Bound). *Given $w \in \mathbb{R}^n$ s.t. $\Phi_{dc}(w) \leq \frac{3}{Cr}$, by choosing a sufficiently large value for $K$, for $w' = w + \eta w_i^* e_i$, we have:*

$$\mathbb{E}_{i\in[n]}[\Phi_{dc}(w')] \leq \Phi_{dc}(w) + \frac{\eta}{2CLnr} .$$

*The assumption $\Phi_{dc}(w) \leq \frac{3}{Cr}$ is justified in the proof of Lemma B.1.*

*Proof.* First we introduce some notation:

Denote $\Phi_{\text{op}}(w) = \min_{\|S\|_F \leq L\|w\|_1}\left(\mu^2 \log\left[F(G_w - S)\right]\right)$, so that $\Phi_{\text{dc}}(w) = \Phi_{\text{op}}(w) + \frac{\|w\|_1}{4CLr}$. Let $G^* = \sum_{i=1}^{n} w_i^* u_i A_i$, and by dRIP Condition 2.3, we know $\exists(L, \frac{1}{K\sqrt{r}})$-norm-decomposition of $G^* = S^* + E^*$, where $\|S^*\|_F \leq L$ and $\|E^*\|_2 \leq \frac{1}{K\sqrt{r}}$. Let $G = \sum_{i=1}^{n} w_i u_i A_i$, and $S =$

argmin$\Phi_{\text{op}}(w)$ so that $\Phi_{\text{op}}(w) = \mu^2 \log[F(G - S)]$, and let $E = G - S$. Let $G' = \sum_{i=1}^{n} w_i' u_i A_i$. Using these notation and $\sum_{i=1}^{n} w_i^* = 1$:

$$\mathbb{E}_{i\in[n]}[\Phi_{\text{dc}}(w')] = \mathbb{E}_{i\in[n]}\left[\Phi_{\text{op}}(w') + \frac{\|w'\|_1}{4CLr}\right] = \mathbb{E}_{i\in[n]}\left[\Phi_{\text{op}}(w') + \frac{\|w\|_1 + \eta w_i^*}{4CLr}\right]$$

$$= \mathbb{E}_{i\in[n]}[\Phi_{\text{op}}(w')] + \frac{\|w\|_1}{4CLr} + \frac{\eta}{4CLnr} \; .$$

We need to show $\mathbb{E}_{i\in[n]}[\Phi_{\text{dc}}(w')] \leq \Phi_{\text{dc}}(w) + \frac{\eta}{2CLnr} = \Phi_{\text{op}}(w) + \frac{\|w\|_1}{4CLr} + \frac{\eta}{2CLnr}$ , equivalently

$$\mathbb{E}_{i\in[n]}[\Phi_{\text{op}}(w')] \leq \Phi_{\text{op}}(w) + \frac{\eta}{4CLnr} \; .$$

Consider $S' = S + \eta w_i^* S^*$. We have $\|S'\|_F \leq \|S\|_F + \eta w_i^* \|S^*\|_F \leq L \cdot \|w\|_1 + \eta w_i^* L = L \cdot \|w'\|_1$, so $S'$ is a valid argument for $\Phi_{\text{op}}(w') = \min_{\|S\|_F \leq L\|w'\|_1} \left(\mu^2 \log\left[F(G' - S)\right]\right)$, therefore $\Phi_{\text{op}}(w') \leq \mu^2 \log[F(G' - S')]$. Let $E' = G' - S' = G + \eta w_i^* u_i A_i - S - \eta w_i^* S^* = E + Z^{(i)}$ where $Z^{(i)} = \eta w_i^* u_i A_i - \eta w_i^* S^*$. Using these and the concavity of the log function, we have

$$\mathbb{E}_{i\in[n]}[\Phi_{\text{op}}(w')] \leq \frac{1}{n}\sum_{i=1}^{n} \mu^2 \log[F(G' - S')] = \frac{1}{n}\sum_{i=1}^{n} \mu^2 \log[F(E + Z^{(i)})]$$

$$\leq \mu^2 \log\left[\frac{1}{n}\sum_{i=1}^{n}\left(F(E + Z^{(i)})\right)\right] \; .$$

It suffices to show

$$\mu^2 \log\left[\frac{1}{n}\sum_{i=1}^{n}\left(F(E + Z^{(i)})\right)\right] \leq \Phi_{\text{op}}(w) + \frac{\eta}{4CLnr} = \mu^2 \log[F(E)] + \frac{1}{4CL}\frac{\eta}{nr} \; .$$

Expanding the left hand side:

$$\mu^2 \log\left[\frac{1}{n}\sum_{i=1}^{n} F(E + Z^{(i)})\right]$$

$$= \mu^2 \log\left[\frac{1}{n}\sum_{i=1}^{n} \text{tr} \exp\left(\frac{E^\top E + Z^{(i)\top}Z^{(i)} + E^\top Z^{(i)} + Z^{(i)\top}E}{\mu^2}\right)\right]$$

$$\leq \mu^2 \log\left[\frac{1}{n}\sum_{i=1}^{n} \text{tr}\left[\exp\left(\frac{E^\top E}{\mu^2}\right) \cdot \exp\left(\frac{Z^{(i)\top}Z^{(i)} + E^\top Z^{(i)} + Z^{(i)\top}E}{\mu^2}\right)\right]\right]$$

$$= \mu^2 \log\left[\text{tr}\left[\exp\left(\frac{E^\top E}{\mu^2}\right) \cdot \frac{1}{n}\sum_{i=1}^{n} \exp\left(\frac{Z^{(i)\top}Z^{(i)} + E^\top Z^{(i)} + Z^{(i)\top}E}{\mu^2}\right)\right]\right]$$

$$\leq \mu^2 \log\left[\text{tr}\exp\left(\frac{E^\top E}{\mu^2}\right) \cdot \left\|\frac{1}{n}\sum_{i=1}^{n} \exp\left(\frac{Z^{(i)\top}Z^{(i)} + E^\top Z^{(i)} + Z^{(i)\top}E}{\mu^2}\right)\right\|_2\right]$$

$$= \mu^2 \log\left[F(E)\right] + \mu^2 \log\left\|\frac{1}{n}\sum_{i=1}^{n} \exp\left(\frac{Z^{(i)\top}Z^{(i)} + E^\top Z^{(i)} + Z^{(i)\top}E}{\mu^2}\right)\right\|_2$$

The first inequality uses Golden-Thompson Inequality (stated as Lemma B.6), and the second inequality follows from Lemma B.4. Finally it suffices to show

$$\mu^2 \log\left\|\frac{1}{n}\sum_{i=1}^{n} \exp\left(\frac{Z^{(i)\top}Z^{(i)} + E^\top Z^{(i)} + Z^{(i)\top}E}{\mu^2}\right)\right\|_2 \leq \frac{1}{4CL}\frac{\eta}{nr} \; .$$

We will use the approximation $\exp(X) \preccurlyeq I + X + X^2$ for symmetric $X$ with $\|X\|_2 \leq 1$. The argument in the exponential satisfies this condition as justified in Claim B.2. We will also use

$\log\left(1+x\right) \le x \ \forall x \ge 0.$

$$\mu^2 \log \left\| \frac{1}{n} \sum_{i=1}^{n} \exp\left( \frac{Z^{(i)\top} Z^{(i)} + E^\top Z^{(i)} + Z^{(i)\top} E}{\mu^2} \right) \right\|_2$$

$$\le \mu^2 \log \left[ \frac{1}{n} \sum_{i=1}^{n} \left\| \exp\left( \frac{Z^{(i)\top} Z^{(i)} + E^\top Z^{(i)} + Z^{(i)\top} E}{\mu^2} \right) \right\|_2 \right]$$

$$\le \mu^2 \log \left[ \frac{1}{n} \sum_{i=1}^{n} \left\| I + \frac{Z^{(i)\top} Z^{(i)} + E^\top Z^{(i)} + Z^{(i)\top} E}{\mu^2} + \frac{(Z^{(i)\top} Z^{(i)} + E^\top Z^{(i)} + Z^{(i)\top} E)^2}{\mu^4} \right\|_2 \right]$$

$$\le \mu^2 \log \left[ \|I\|_2 + \left\| \frac{1}{n} \sum_{i=1}^{n} \frac{Z^{(i)\top} Z^{(i)}}{\mu^2} \right\|_2 + \left\| \frac{1}{n} \sum_{i=1}^{n} \frac{E^\top Z^{(i)} + Z^{(i)\top} E}{\mu^2} \right\|_2 \right.$$
$$\left. + \left\| \frac{1}{n} \sum_{i=1}^{n} \frac{(Z^{(i)\top} Z^{(i)} + E^\top Z^{(i)} + Z^{(i)\top} E)^2}{\mu^4} \right\|_2 \right]$$

$$\le \mu^2 \log \left[ \|I\|_2 + \left\| \frac{1}{n} \sum_{i=1}^{n} \frac{Z^{(i)\top} Z^{(i)}}{\mu^2} \right\|_2 + \left\| \frac{1}{n} \sum_{i=1}^{n} \frac{E^\top Z^{(i)} + Z^{(i)\top} E}{\mu^2} \right\|_2 \right.$$
$$\left. + \left\| \frac{1}{n} \sum_{i=1}^{n} \frac{2(Z^{(i)\top} Z^{(i)})^2}{\mu^4} \right\|_2 + \left\| \frac{1}{n} \sum_{i=1}^{n} \frac{2(E^\top Z^{(i)} + Z^{(i)\top} E)^2}{\mu^4} \right\|_2 \right]$$

$$\le \mu^2 \left\| \frac{1}{n} \sum_{i=1}^{n} \frac{Z^{(i)\top} Z^{(i)}}{\mu^2} \right\|_2 + \mu^2 \left\| \frac{1}{n} \sum_{i=1}^{n} \frac{E^\top Z^{(i)} + Z^{(i)\top} E}{\mu^2} \right\|_2$$
$$+ 2\mu^2 \left\| \frac{1}{n} \sum_{i=1}^{n} \frac{(Z^{(i)\top} Z^{(i)})^2}{\mu^4} \right\|_2 + 2\mu^2 \left\| \frac{1}{n} \sum_{i=1}^{n} \frac{(E^\top Z^{(i)} + Z^{(i)\top} E)^2}{\mu^4} \right\|_2$$

$$= \left\| \frac{1}{n} \sum_{i=1}^{n} Z^{(i)\top} Z^{(i)} \right\|_2 + \left\| \frac{1}{n} \sum_{i=1}^{n} E^\top Z^{(i)} + Z^{(i)\top} E \right\|_2$$
$$+ \frac{2}{\mu^2} \left\| \frac{1}{n} \sum_{i=1}^{n} (Z^{(i)\top} Z^{(i)})^2 \right\|_2 + \frac{2}{\mu^2} \left\| \frac{1}{n} \sum_{i=1}^{n} (E^\top Z^{(i)} + Z^{(i)\top} E)^2 \right\|_2$$

These four terms are bounded by Claims B.3,B.4, B.5 and B.6 respectively, notice that the second term dominates the first and the third, and the forth term dominates the second. So finally we have:

$$\mu^2 \log \left\| \frac{1}{n} \sum_{i=1}^{n} \exp\left( \frac{Z^{(i)\top} Z^{(i)} + E^\top Z^{(i)} + Z^{(i)\top} E}{\mu^2} \right) \right\|_2$$
$$\le \left( \frac{3 \times 4}{\sqrt{C} K} + \frac{96 L^2}{K} \right) \frac{\eta}{nr}$$
$$\le \frac{97 L^2}{K} \frac{\eta}{nr}$$
$$= \frac{1}{4CL} \frac{\eta}{nr} \text{ , with choice of } K = 388 C L^3 = O(L^3) . \qquad \square$$

**Lemma B.4.** *If* $0 \preccurlyeq A$, *then* $\mathrm{tr}(AB) \le \mathrm{tr}(A) \cdot \|B\|_2$ .

*Proof.* Since $0 \preccurlyeq A$, $A = \sum_j \sigma_j u_j u_j^\top$ with $\sigma_j \geq 0$.

$$\text{tr}(AB) = \text{tr}\left(\sum_j \sigma_j u_j u_j^\top B\right) = \sum_j \sigma_j \text{tr}(u_j u_j^\top B) = \sum_j \sigma_j u_j^\top B u_j$$
$$\leq \sum_j \sigma_j \cdot \|B\|_2$$
$$= \text{tr}(A) \cdot \|B\|_2 . \qquad \square$$

**Lemma B.5.** $(A + B)^\top (A + B) \preccurlyeq 2A^\top A + 2B^\top B$, *and* $\left[(A + B)^\top (A + B)\right]^2 \preccurlyeq 8(A^\top A)^2 + 8(B^\top B)^2$.

*Proof.*

$$2A^\top A + 2B^\top B - (A + B)^\top (A + B) = A^\top A + B^\top B - A^\top B - B^\top A$$
$$= (A - B)^\top (A - B)$$
$$\succcurlyeq 0 .$$

$$\left[(A + B)^\top (A + B)\right]^2 \preccurlyeq (2A^\top A + 2B^\top B)^2$$
$$\preccurlyeq 2[2(A^\top A)]^2 + 2[2(B^\top B)]^2$$
$$\preccurlyeq 8(A^\top A)^2 + 8(B^\top B)^2 . \qquad \square$$

The following claims were used in Lemma B.3. Recall that $\mathcal{A}$ satisfies dRIP Condition 2.3, $u = \mathcal{A}[V] \in \mathbb{R}^n$ for some $V \in \mathbb{R}^{d_1 \times d_2}$ satisfying $\|V\|_F \in [\frac{1}{4}, 1]$, $\|V\|_* \leq 2\sqrt{2r}$, $Z^{(i)} = \eta w_i^*(u_i A_i - S^*)$, and $\Phi_{\text{dc}}(w) \leq \frac{3}{Cr}$ by assumption of Lemma B.3. We have the following:

$\|A_i\|_2 \leq \rho$ by boundedness property of dRIP Condition 2.3,

$|u_i| = |\langle A_i, V\rangle| \leq \|A_i\|_2 \|V\|_* \leq \rho 2\sqrt{2r} \leq L\sqrt{r}\rho$ assuming $2\sqrt{2} \leq L$,

$\|S^*\|_2 \leq \|S^*\|_F \leq L, \|E^*\|_2 \leq \dfrac{1}{K\sqrt{r}}$ by decomposition property of dRIP Condition 2.3,

$\|E\|_2^2 \leq \Phi_{\text{op}}(w) \leq \Phi_{\text{dc}}(w) \leq \dfrac{3}{Cr}$ by Claim B.1.

**Claim B.2.** $\left\|\frac{Z^{(i)\top} Z^{(i)} + E^\top Z^{(i)} + Z^{(i)\top} E}{\mu^2}\right\|_2 \leq 1$.

*Proof.*

$$\left\|Z^{(i)\top} Z^{(i)}\right\|_2 = \eta^2 w_i^{*2} \left\|(u_i A_i - S^*)^\top (u_i A_i - S^*)\right\|_2$$
$$\leq 2\eta^2 w_i^{*2} \left(u_i^2 \left\|A_i^\top A_i\right\|_2 + \left\|S^{*\top} S^*\right\|_2\right) \text{ (Lemma B.5)}$$
$$\leq 2\eta^2 w_i^{*2}(L^2 r\rho^4 + L^2)$$
$$\leq 4\eta^2 w_i^{*2} L^2 r\rho^4$$

$$\left\|E^\top Z^{(i)} + Z^{(i)\top} E\right\|_2 \leq 2\left\|E^\top Z^{(i)}\right\|_2 \leq 2\|E\|_2 \cdot \left\|Z^{(i)}\right\|_2$$
$$= 2\eta w_i^* \|E\|_2 \cdot \|u_i A_i - S^*\|_2$$
$$\leq 2\eta w_i^* \|E\|_2 \cdot (|u_i| \|A_i\|_2 + \|S^*\|_2)$$
$$\leq 2\eta w_i^* \frac{2}{\sqrt{Cr}}(L\sqrt{r}\rho^2 + L)$$
$$\leq 8\eta w_i^* \frac{L\rho^2}{\sqrt{C}}$$

Putting them together:

$$\left\| \frac{Z^{(i)\top}Z^{(i)} + E^\top Z^{(i)} + Z^{(i)\top}E}{\mu^2} \right\|_2 \leq \frac{1}{\mu^2}\left( \left\|Z^{(i)\top}Z^{(i)}\right\|_2 + \left\|E^\top Z^{(i)} + Z^{(i)\top}E\right\|_2 \right)$$

$$\leq Cr\log d \cdot (4\eta^2 w_i^{*2}L^2 r\rho^4 + 8\eta w_i^* \frac{L\rho^2}{\sqrt{C}})$$

$$\leq \frac{16\sqrt{C}L}{K} \leq 1 \text{ with sufficiently large } K . \qquad \square$$

**Claim B.3.** $\left\|\frac{1}{n}\sum_{i=1}^n Z^{(i)\top}Z^{(i)}\right\|_2 \leq \frac{4L^2}{K\log d} \cdot \frac{\eta}{nr}$ .

*Proof.*

$$\sum_{i=1}^n \left\|Z^{(i)\top}Z^{(i)}\right\|_2 = \sum_{i=1}^n \eta^2 w_i^{*2}\left\|(u_iA_i - S^*)^\top(u_iA_i - S^*)\right\|_2$$

$$\leq 2\sum_{i=1}^n \eta^2 w_i^{*2}\left(u_i^2\left\|A_i^\top A_i\right\|_2 + \left\|S^{*\top}S^*\right\|_2\right)$$

$$\leq 2\eta \sum_{i=1}^n \eta w_i^*(w_i^*u_i^2\rho^2 + w_i^*L^2)$$

$$\leq 2\eta \frac{1}{Kr\rho^2\log d}(L\rho^2 + L^2)$$

$$\leq \frac{4L^2}{K\log d}\frac{\eta}{r}$$

$$\implies \left\|\frac{1}{n}\sum_{i=1}^n Z^{(i)\top}Z^{(i)}\right\|_2 \leq \frac{1}{n}\sum_{i=1}^n \left\|Z^{(i)\top}Z^{(i)}\right\|_2$$

$$\leq \frac{4L^2}{K\log d} \cdot \frac{\eta}{nr} . \qquad \square$$

**Claim B.4.** $\left\|\frac{1}{n}\sum_{i=1}^n E^\top Z^{(i)} + Z^{(i)\top}E\right\|_2 \leq \frac{4}{\sqrt{C}K} \cdot \frac{\eta}{nr}$ .

*Proof.*

$$\left\|\frac{1}{n}\sum_{i=1}^n E^\top Z^{(i)} + Z^{(i)\top}E\right\|_2 = \frac{1}{n}\left\|E^\top \sum_{i=1}^n Z^{(i)} + \sum_{i=1}^n Z^{(i)\top}E\right\|_2$$

$$\leq \frac{2}{n}\left\|E^\top \sum_{i=1}^n Z^{(i)}\right\|_2$$

$$\leq \frac{2}{n}\left\|E\right\|_2 \cdot \left\|\sum_{i=1}^n Z^{(i)}\right\|_2$$

$$= \frac{2}{n}\left\|E\right\|_2 \cdot \left\|\eta G^* - \eta S^*\right\|_2$$

$$\leq \frac{2}{n}\left\|E\right\|_2 \cdot \eta\left\|E^*\right\|_2$$

$$\leq \frac{2}{n} \cdot \frac{2}{\sqrt{C}r} \cdot \frac{\eta}{K\sqrt{r}}$$

$$= \frac{4}{\sqrt{C}K}\frac{\eta}{nr} . \qquad \square$$

**Claim B.5.** $\frac{2}{\mu^2}\left\|\frac{1}{n}\sum_{i=1}^n (Z^{(i)\top}Z^{(i)})^2\right\|_2 \leq O\left(\frac{CL^4}{K^3 r\rho^2\log^2 d}\right) \cdot \frac{\eta}{nr}$ .

*Proof.*

$$\left\|\sum_{i=1}^{n}(Z^{(i)\top}Z^{(i)})^2\right\|_2 \le 8\sum_{i=1}^{n}\eta^4 w_i^{*4}\left(u_i^4\cdot\|(A_i^\top A_i)^2\|_2+\|(S^{*\top}S^*)^2\|_2\right) \text{ (Lemma B.5)}$$

$$\le 8\eta\sum_{i=1}^{n}\eta^3 w_i^{*2}(w_i^{*2}u_i^4\rho^4+w_i^{*2}L^4)$$

$$\le 8\eta\cdot\frac{1}{K^3 r^3\rho^6\log^3 d}\left[\sum_{i=1}^{n}w_i^{*2}\rho^4 u_i^4+\sum_{i=1}^{n}w_i^{*2}L^4\right]$$

$$\le 8\eta\cdot\frac{1}{K^3 r^3\rho^6\log^3 d}\left[\left(\sum_{i=1}^{n}w_i^*\rho^2 u_i^2\right)^2+\left(\sum_{i=1}^{n}w_i^* L^2\right)^2\right]$$

$$\le 8\eta\cdot\frac{1}{K^3 r^3\rho^6\log^3 d}(\rho^4 L^2+L^4)$$

$$\le\frac{16L^4}{K^3 r^2\rho^2\log^3 d}\cdot\frac{\eta}{r}$$

$$\le O\left(\frac{L^4}{K^3 r^2\rho^2\log^3 d}\right)\cdot\frac{\eta}{r}\;.$$

$$\implies\frac{2}{\mu^2}\left\|\frac{1}{n}\sum_{i=1}^{n}(Z^{(i)\top}Z^{(i)})^2\right\|_2\le O\left(\frac{CL^4}{K^3 r\rho^2\log^2 d}\right)\cdot\frac{\eta}{nr} \qquad\square$$

**Claim B.6.** $\frac{2}{\mu^2}\left\|\frac{1}{n}\sum_{i=1}^{n}(E^\top Z^{(i)}+Z^{(i)\top}E)^2\right\|_2\le\frac{96L^2}{K}\cdot\frac{\eta}{nr}$ .

*Proof.*

$$\frac{2}{\mu^2}\left\|\frac{1}{n}\sum_{i=1}^{n}(E^\top Z^{(i)}+Z^{(i)\top}E)^2\right\|_2\le\frac{2}{\mu^2}\frac{1}{n}\sum_{i=1}^{n}\left\|(E^\top Z^{(i)}+Z^{(i)\top}E)^2\right\|_2$$

$$\le\frac{2}{\mu^2}\frac{1}{n}\sum_{i=1}^{n}4\left\|E^\top Z^{(i)}Z^{(i)\top}E\right\|_2$$

$$\le\frac{2}{\mu^2}\frac{1}{n}\sum_{i=1}^{n}4\|E\|_2^2\left\|Z^{(i)\top}Z^{(i)}\right\|_2$$

$$\le\frac{8}{\mu^2}\|E\|_2^2\cdot\frac{1}{n}\sum_{i=1}^{n}\left\|Z^{(i)\top}Z^{(i)}\right\|_2$$

$$\le\frac{8}{\mu^2}\cdot\frac{3}{Cr}\cdot\frac{4L^2}{K\log d}\frac{\eta}{nr} \text{ (Claim B.3)}$$

$$=\frac{96L^2}{K}\cdot\frac{\eta}{nr} \qquad\square$$

**Lemma B.6** (Golden–Thompson Inequality [Tho65]). *For two $n\times n$ Hermitian matrices $A$ and $B$:*

$$\text{tr}\left(\exp(A+B)\right)\le\text{tr}\left(\exp(A)\exp(B)\right)\;.$$

**Lemma 5.4** (Algorithm 3 Running Time). *Algorithm 3 with failure probability $\delta$ runs in time $O(nd^\omega\,\text{polylog}\,(d)\log\frac{1}{\delta}r\rho^2)$ .*

*Proof.* Algorithm 3 has a nested for loop that's repeated for $N'\times N=O(n\log d\log\frac{1}{\delta}r\rho^2)$ times. The major step in the loop is line 11: $s_t\leftarrow\arg\max_{s\in[0,\eta]}\Phi(w_t+se_i)$, which is equivalent to $\arg\min_{s\in[0,\eta]}Cr\Phi_{\text{op}}(w+se_i)+\frac{s}{4L}-su_i^2$. Recall that $\Phi_{\text{op}}(w)=\min\limits_{\|S\|_F\le L\|w\|_1}\left(\mu^2\log\left[F(G_w-S)\right]\right)$ where $F(E)=\text{tr}\exp\left(\frac{E^\top E}{\mu^2}\right)$. Note that $\mu^2\log\left[F(E)\right]$ is convex in $E$.

First we show that $\Phi_{\text{op}}(w)$ is convex in $w$, i.e., given $w_1, w_2$, $\Phi_{\text{op}}\left(\frac{1}{2}(w_1 + w_2)\right) \leq \frac{1}{2}\left(\Phi_{\text{op}}(w_1) + \Phi_{\text{op}}(w_2)\right)$.

Let $w_3 = \frac{1}{2}(w_1 + w_2)$, and $G_k = \sum_{i=1}^{n}(w_k)_i u_i A_i$ for $k = 1, 2, 3$. Suppose $S_1, S_2$ attain the minimum for $\Phi_{\text{op}}(w_1), \Phi_{\text{op}}(w_2)$ respectively, i.e., $\Phi_{\text{op}}(w_1) = \mu^2 \log\left[F(G_1 - S_1)\right]$ and $\Phi_{\text{op}}(w_2) = \mu^2 \log\left[F(G_2 - S_2)\right]$.

Let $S_3 = \frac{1}{2}(S_1 + S_2)$. Notice that $G_3 = \frac{1}{2}(G_1 + G_2)$, so $G_3 - S_3 = \frac{1}{2}(G_1 - S_1 + G_2 - S_2)$. Since $\|S_3\|_F \leq \frac{1}{2}(\|S_1\|_F + \|S_2\|_F) \leq \frac{1}{2}L(\|w_1\|_1 + \|w_2\|_1) = L\|w_3\|_1$, $S_3$ is a valid argument for $\Phi_{\text{op}}(w_3)$, therefore:

$$\Phi_{\text{op}}(w_3) \leq \mu^2 \log\left[F(G_3 - S_3)\right] \leq \frac{1}{2}\left(\mu^2 \log\left[F(G_1 - S_1)\right] + \mu^2 \log\left[F(G_2 - S_2)\right]\right)$$

$$= \frac{1}{2}\left(\Phi_{\text{op}}(w_1) + \Phi_{\text{op}}(w_2)\right).$$

Line 11 is equivalent to minimizing $Cr\Phi_{\text{op}}(w + se_i) + \frac{s}{4L} - su_i^2$, which is convex in $s$ for a fixed $w$, over a bounded interval $[0, \eta]$, so the minimization needs to evaluate $\Phi_{\text{op}}(w + se_i)$ for $O(\text{polylog}(d))$ different values of $s$. Evaluating $\Phi_{\text{op}}$ is also a minimization which can be solved by computing SVD on $G_w$ and evaluating $F(G_w - S)$ in time $O(d^\omega)$ for $\text{polylog}(d)$ various constructions of $S$. Overall finding the optimal value of $s$ takes time $O(d^\omega \cdot \text{polylog}(d))$, and the algorithm's total running time is $O(nd^\omega \text{polylog}(d) \log \frac{1}{\delta} r\rho^2)$. $\qquad\square$

## C   Omitted proofs: From wRIP to dRIP condition

Here we show that the dRIP Condition 2.3 is implied by the wRIP Condition 2.2, given proper choices of parameters within a constant factor. Notice that in the wRIP condition, we have a low-rank constraint on the input matrix, i.e., $\text{rank}(X) \leq r$, and in dRIP we have a norm constraint instead, i.e., $\|V\|_F \in [\frac{1}{4}, 1]$, $\|V\|_* \leq 2\sqrt{2r}$. To make use of the wRIP condition of $\mathcal{A}$, we will decompose $V$ into low-rank matrices, so that wRIP condition applies to each of the low-rank matrices. Though the rank of $V$ is arbitrary, we can still upper bound its numerical rank based on the norm constraint.

First we will introduce a low-rank decomposition, and an upper bound on the sum of their Frobenius norms. This is the matrix version of the shelling-decomposition in Lemma 15 for vectors in [KLL$^+$23].

**Lemma C.1** (Low-rank Decomposition). *Given $V \in \mathbb{R}^{d_1 \times d_2}$ with $\text{Rank}_{\text{n}}(V) = \frac{\|V\|_*^2}{\|V\|_F^2} = \nu$, and let $V = \sum \sigma_i u_i v_i^\top$ be its SVD with $\sigma_i$ in descending order. Decompose $V$ into sum of rank-$r$ matrices, i.e., write $V = \sum_{\ell=1}^{\ell=k} V^{(\ell)}$ where $V^{(\ell)} = \sum_{i=(\ell-1)r+1}^{i=\ell r} \sigma_i u_i v_i^\top$. Then we have: $\sum_{\ell=2}^{k}\left\|V^{(\ell)}\right\|_F \leq \sqrt{\frac{\nu}{r}}\|V\|_F$.*

*Proof.* Note that $V^{(1)}$ is the rank-$r$ approximation of $V$, and $V^{(\ell)}$'s are constructed using disjoint singular values and vectors in groups of size $r$, and are orthogonal to each other.

Denote $\sigma_i\left(V^{(\ell)}\right)$ as the $i^{\text{th}}$ largest singular value of $V^{(\ell)}$.

$$\left\|V^{(\ell+1)}\right\|_F \leq \sqrt{r} \cdot \sigma_1\left(V^{(\ell+1)}\right) \leq \sqrt{r} \cdot \sigma_r\left(V^{(\ell)}\right) \leq \sqrt{r} \cdot \frac{\left\|V^{(\ell)}\right\|_*}{r},$$

$$\sum_{\ell=2}^{k}\left\|V^{(\ell)}\right\|_F \leq \frac{\sqrt{r}}{r} \cdot \sum_{\ell=1}^{k-1}\left\|V^{(\ell)}\right\|_* \leq \frac{\sqrt{r}}{r} \cdot \sum_{\ell=1}^{k}\left\|V^{(\ell)}\right\|_* = \frac{\sqrt{r}}{r} \cdot \|V\|_*,$$

$$\frac{\|V\|_*^2}{\|V\|_F^2} \leq \nu \implies \sum_{\ell=2}^{k}\left\|V^{(\ell)}\right\|_F \leq \sqrt{\frac{\nu}{r}}\|V\|_F. \qquad\square$$

Now we are ready to prove Lemma 5.1, which states that wRIP implies dRIP condition. The proof uses similar techniques as in the second part of Lemma 17 [KLL$^+$23] for vector recovery.

**Lemma 5.1** (wRIP $\implies$ dRIP). *If $\mathcal{A}$ satisfies wRIP Condition 2.2 with parameters $r', L', \rho$, then $\mathcal{A}$ satisfies the dRIP Condition 2.3 with parameters $L, K, r, \rho$ such that $L = \Theta(L')$, $r = \Theta(r')$, and some constant $K \geq 1$.*

*Proof.* **Boundedness property** is satisfied by assumption $\|A_i\|_2 \le \rho \,\forall i$ .

**Isometry property**:

Consider $V \in \mathbb{R}^{d_1 \times d_2}$ $s.t.$ $\|V\|_F \in [\frac{1}{4}, 1]$, $\|V\|_* \le 2\sqrt{2r}$. Need to show $\frac{1}{L} \le \sum_{i=1}^n w_i^* \langle A_i, V \rangle^2 \le L$ .

Let $L = 25L'$, $K \ge 1$ and $r = \frac{r'}{12800L^2K^2}$ .

$\nu = \mathrm{Rank}_n(V) = \frac{\|V\|_*^2}{\|V\|_F^2} \le 128r$. By Lemma C.1, decompose $V$ into rank-$r'$ matrices so that we can apply the $(r', L')$-wRIP property of $\mathcal{A}$.

$$\sum_{\ell=2}^k \left\|V^{(\ell)}\right\|_F \le \sqrt{\frac{\nu}{r'}}\|V\|_F \le \frac{1}{10LK}\|V\|_F \le \frac{1}{10}\|V\|_F \,,$$

$$\|V\|_F \ge \left\|V_{(r')}\right\|_F = \left\|V^{(1)}\right\|_F = \left\|V - \sum_{\ell=2}^k V^{(\ell)}\right\|_F \ge \|V\|_F - \sum_{\ell=2}^k \left\|V^{(\ell)}\right\|_F \ge \frac{9}{10}\|V\|_F \,.$$

Let $B_i = \sqrt{w_i^*}A_i$, so that $\sum_{i=1}^n w_i^* \langle A_i, V \rangle^2 = \sum_{i=1}^n \langle B_i, V \rangle^2 = \left\|\sum_{i=1}^n \langle B_i, V \rangle e_i\right\|_2^2$ .

Lower bound:

$$\left\|\sum_{i=1}^n \langle B_i, V \rangle e_i\right\|_2 \ge \left\|\sum_{i=1}^n \langle B_i, V^{(1)} \rangle e_i\right\|_2 - \left\|\sum_{i=1}^n \langle B_i, \sum_{\ell=2}^k V^{(\ell)} \rangle e_i\right\|_2$$

$$\ge \left\|\sum_{i=1}^n \langle B_i, V^{(1)} \rangle e_i\right\|_2 - \sum_{\ell=2}^k \left\|\sum_{i=1}^n \langle B_i, V^{(\ell)} \rangle e_i\right\|_2$$

$$= \sqrt{\sum_{i=1}^n \langle B_i, V^{(1)} \rangle^2} - \sum_{\ell=2}^k \sqrt{\sum_{i=1}^n \langle B_i, V^{(\ell)} \rangle^2}$$

$$\ge \sqrt{\frac{1}{L'} \cdot \left\|V^{(1)}\right\|_F^2} - \sum_{\ell=2}^k \sqrt{L' \cdot \left\|V^{(\ell)}\right\|_F^2}$$

$$\ge \frac{0.9}{\sqrt{L'}} \cdot \|V\|_F - \frac{0.1}{L}\sqrt{L'} \cdot \|V\|_F$$

$$= \frac{4.5}{\sqrt{L}} \cdot \|V\|_F - \frac{0.02}{\sqrt{L}} \cdot \|V\|_F \,.$$

Taking the square: $\sum_{i=1}^n w_i^* \langle A_i, V \rangle^2 \ge \frac{4.48^2}{L} \cdot \|V\|_F^2 \ge \frac{4.48^2}{16L} \ge \frac{1}{L}$ .

Upper bound:

$$\left\|\sum_{i=1}^n \langle B_i, V \rangle e_i\right\|_2 \le \left\|\sum_{i=1}^n \langle B_i, V^{(1)} \rangle e_i\right\|_2 + \sum_{\ell=2}^k \left\|\sum_{i=1}^n \langle B_i, V^{(\ell)} \rangle e_i\right\|_2$$

$$= \sqrt{\sum_{i=1}^n \langle B_i, V^{(1)} \rangle^2} + \sum_{\ell=2}^k \sqrt{\sum_{i=1}^n \langle B_i, V^{(\ell)} \rangle^2}$$

$$\le \sqrt{L'} \cdot \left\|V^{(1)}\right\|_F + \sum_{\ell=2}^k \sqrt{L'} \cdot \left\|V^{(\ell)}\right\|_F$$

$$\le \sqrt{L'} \cdot \|V\|_F^2 + \frac{0.1\sqrt{L'}}{L} \cdot \|V\|_F$$

$$= \frac{\sqrt{L}}{5} \cdot \|V\|_F + \frac{0.02}{\sqrt{L}} \cdot \|V\|_F \,.$$

Taking the square: $\sum_{i=1}^n w_i^* \langle A_i, V \rangle^2 \le \frac{L}{25} \cdot \|V\|_F^2 + \frac{0.02^2}{L} \cdot \|V\|_F^2 + 0.008\|V\|_F^2 \le L \cdot \|V\|_F^2 \le L$ .

Combining the lower bound and upper bound: $\frac{1}{L} \le \sum_{i=1}^n w_i^* \langle A_i, V \rangle^2 \le L$ .

**Decomposition property**:

Let $S = G_{(r')}$, the rank-$r'$ approximation of $G = \sum_{i=1}^{n} w_i^* \langle A_i, V \rangle A_i$. Let $E = G - S$. Suffices to show $\|S\|_F \le L$ and $\|E\|_2 \le \frac{1}{K\sqrt{r}}$. We have

$$\|S\|_F^2 = \langle S, S \rangle = \langle G, S \rangle = \langle \sum_{i=1}^{n} w_i^* \langle A_i, V \rangle A_i, S \rangle = \langle \sum_{i=1}^{n} \langle B_i, V \rangle B_i, S \rangle = \sum_{i=1}^{n} \langle B_i, V \rangle \langle B_i, S \rangle$$

$$= \langle \sum_{i=1}^{n} \langle B_i, V \rangle e_i, \sum_{j=1}^{n} \langle B_j, S \rangle e_j \rangle \le \left\| \sum_{i=1}^{n} \langle B_i, V \rangle e_i \right\|_2 \cdot \left\| \sum_{i=1}^{n} \langle B_i, S \rangle e_i \right\|_2$$

$$= \sqrt{\sum_{i=1}^{n} \langle B_i, V \rangle^2} \cdot \sqrt{\sum_{i=1}^{n} \langle B_i, S \rangle^2} = \sqrt{\sum_{i=1}^{n} w_i^* \langle A_i, V \rangle^2} \cdot \sqrt{\sum_{i=1}^{n} w_i^* \langle A_i, S \rangle^2}$$

$$\le \sqrt{L} \cdot \sqrt{L'} \cdot \|S\|_F \le \frac{L}{5} \|S\|_F$$

which implies $\|S\|_F \le \frac{L}{5} \le L$, and consequently,

$$\|E\|_2 = \sigma_{r'+1}(G) \le \sigma_{r'}(G) = \sigma_{r'}(S) \le \sqrt{\frac{\|S\|_F^2}{r'}} \le \frac{L}{5\sqrt{r'}} = \frac{L}{400\sqrt{2}LK\sqrt{r}} \le \frac{1}{K\sqrt{r}} \ . \quad \square$$