# OpenReview forum: "Robust Matrix Sensing in the Semi-Random Model"
_NeurIPS.cc/2023/Conference — NeurIPS 2023 poster_

### Official Review · Reviewer_kSWp · 2023-06-09

**Soundness:** 3 good
**Presentation:** 3 good
**Contribution:** 3 good
**Rating:** 6
**Confidence:** 4

**Summary:**

This paper studies the problem of low-rank matrix recovery for semi-random measurement matrices. This model allows for a mix of RIP matrices and arbitrarily chosen matrices. Using an iterative re-weighted approach, the authors demonstrate provable recovery using concepts from sparse recovery.

**Strengths:**

The paper does a great job and connecting the sparse recovery framework to the setting studied. Although this indeed is not a new observation, it is discussed and explained well here. The paper is overall written well and the results are solid and robust. The theoretical guarantees are clearly stated and give precise recovery guarantees in this framework.

**Weaknesses:**

The semi-random model is interesting mathematically but could be motivated better practically within the paper. The paper would be stronger if it included experimental results that supported the theory.

**Questions:**

Do the authors believe their results can apply to combinations of sparse and low-rank matrices?

**Limitations:**

Future work is discussed.

---

> ### Author Rebuttal · Authors · 2023-08-09
>
> We thank the reviewer for the thoughtful comments and positive feedback.
>
> On “combinations of sparse and low-rank matrices”: If we understand correctly, the reviewer is referring to the problem of sensing a matrix M in the semi-random model, in which the underlying matrix M can be decomposed as M = A + B where A is low-rank (and incoherent) and B is sparse. One possible approach is to maintain low-rank matrices A_t and B_t in each iteration, run alternating weighted gradient descent using the weight oracles proposed in [KLL+23] and in our paper, and show that A_t and B_t converge to the ground truth matrices. While we have not explored this setting on a technical level, this is an exciting avenue for future work and we thank the reviewer for pointing us in this direction.
>
> [KLL+23] Jonathan A. Kelner, Jerry Li, Allen Liu, Aaron Sidford, Kevin Tian. Semi-Random Sparse Recovery in Nearly-Linear Time. COLT 2023.

---

> > ### Comment · Reviewer_kSWp · 2023-08-11
> >
> > Thank you for your responses.

---

### Official Review · Reviewer_qNDU · 2023-06-29

**Soundness:** 3 good
**Presentation:** 4 excellent
**Contribution:** 3 good
**Rating:** 5
**Confidence:** 3

**Summary:**

This paper study the matrix sensing problem (or low-rank matrix recovery) in the so-called semi-random model. The semi-random model stems from the fact that an unknown set of sensing matrices satisfy the Restricted Isometry Properties (RIP) while the remaining set are chosen adversarily. This model has the advantage to design methods that are robust to noise or adversarial examples. The authors design design a descent-style algorithm that iteratively reweights the input to recover the ground truth matrix. They extend a previous work that tackle the sparse vector recovery problem in the semi-random model, to matrix low-rank estimation problem.

**Strengths:**

- The work tackles a very important theoretical problem in Machine learning that have direct implications on designing practical algorithm for robustness in the recovery of low-rank signals.
- The paper generalize existing work [KL22+] from a theoretical standpoint.

**Weaknesses:**

- Lack of empirical evaluation of the main results on at least synthetic data
- Some of the theoretical claims have not been proved in the paper : 1- The authors claim that their work improves correctness and accuracy of existing non-convex relaxations [BNS16] 2- On the [BNS16] method, there are bad local minima when considering the semi-random model
- The authors say that one of the drawback of convex solutions are their prohibitive time complexities. But their results show that they achieve comparable run-time as existing convex solutions


**Questions:**

Q1- Some of the theoretical claims have not been proved in the paper :
1- The authors claim that their work improves correctness and accuracy of existing non-convex relaxations [BNS16]
Could the authors please provide a proof of the statement in the appendix?
2- On the [BNS16] method, there are bad local minima when considering the semi-random model
Could the authors please provide a proof of the statement in the appendix?

Q2- The authors say that one of the drawback of convex solutions are their prohibitive time complexities. But their results show that they achieve comparable run-time as existing convex solutions. Could the authors please elaborate more on this?

Q3- Could the authors please provide a small section of empirical evidence of their results, at least on synthetic data for a low-rank matrix recovery problem under adversary addition to their input, showing the gain of their method (time vs accuracy) over existing approaches?

Q4- Combining the current analysis with the work [KL22+], to which extent the analysis of estimation of SPARSE and LOW-RANK matrix in the semi-random model doable?

Q5- Could you please add the Limitations of the paper, as required in the guidelines?

I would raise my score if the questions Q1-Q3 above are properly addressed.

**Limitations:**

The authors have not provided the limitations of their work in the manuscript. Could they please add the limitations in the rebuttal?

---

> ### Author Rebuttal · Authors · 2023-08-09
>
> We thank the reviewer for the thoughtful comments and positive feedback.
>
> Regarding bad local optima in the semi-random model (Q1, Part 2): Section 5 of [BNS16] proved that bad local minima exist if the sensing matrices do not satisfy RIP. We mentioned this in Lines 74-77 of our paper. More precisely in our setting, a semi-random adversary can add an arbitrary number of sensing matrices (provided that the linear measurements are correct), so we can have 1% of the sensing matrices satisfying RIP and use the remaining 99% to implement the counter-examples in [BNS16]. We will discuss this in more detail, and we will include the counter-examples in the Appendix to be self-contained.
>
> On comparison with existing non-convex approaches such as [BNS16] (Q1, Part 1): We intended to say that existing non-convex approaches can get stuck in bad local optima in the semi-random model, while our algorithm is guaranteed to converge to a global optimum. We agree with the reviewer that the phrase “improves the correctness and accuracy” should be made more precise. We will clarify this.
>
> Regarding our runtime being comparable to convex approaches (Q2): We focused on nonconvex approaches because they have many advantages in practice. For example, gradient descent is conceptually much simpler than state-of-the-art semidefinite program (SDP) solvers, and it can converge in much fewer iterations predicted by theory, while the runtime of cutting-plane and interior-point methods are often close to their theoretical bounds. More importantly, despite having comparable asymptotic runtime as existing convex approaches, we believe our work will serve as a starting point for the discovery of more practical and provably robust algorithms for semi-random matrix sensing.
>
> Regarding experiments (Q3): We believe that our results are substantial even without experiments, and we hope that our theoretical results are judged based on their merits. Our main contribution is to pose and study the problem of semi-random matrix sensing, and to provide descent-style algorithms with good asymptotic runtimes. We did not include experiments because our focus is not to obtain the fastest possible algorithm for semi-random matrix sensing, and experiments sometimes divert readers' attention toward such metrics. That said, we acknowledge that experimental evaluation is important and a fruitful direction for future work.
>
> On recovering sparse and low-rank matrices (Q4): If the ground-truth matrix M is sparse, we can view M as a vector and use [KLL+23] to recover M in the semi-random model. Consequently, when M is sparse *and* low-rank, one can choose between our algorithm and [KLL+23]. The two algorithms have different sample complexities and running times, and the choice should be made depending on the rank and sparsity parameters. A potentially better approach is to integrate the two algorithms, e.g., by running weighted gradient descent and projecting onto the set of matrices with small nuclear norm (as in our paper) and small entrywise L1 norm (as in [KLL+23]), which is an intriguing direction for future work.
>
> On a separate Limitations section (Q5): We would like to bring to the reviewer’s attention that a separate "Limitations" section is not mandatory according to this year’s CFP and Paper Checklist. We stated all assumptions very precisely (e.g., Theorem 3.1) and our algorithm will succeed under these assumptions.
>
>
> [BNS16] Srinadh Bhojanapalli, Behnam Neyshabur, Nathan Srebro. Global Optimality of Local Search for Low Rank Matrix Recovery. NeurIPS 2016.
>
> [KLL+23] Jonathan A. Kelner, Jerry Li, Allen Liu, Aaron Sidford, Kevin Tian. Semi-Random Sparse Recovery in Nearly-Linear Time. COLT 2023.

---

> > ### Comment · Reviewer_qNDU · 2023-08-12
> >
> > Thanks to the authors for clarifications on Q1-Q2. As for the modifications promised by the authors, it would be better to put them in color so it is easier to follow-up?
> > I still believe that some experiments on synthetic data would be good for completeness of the paper, although the focus of the paper is theoretical. Or at least have a small section describing practical benefits of the approach.

---

### Official Review · Reviewer_PQDV · 2023-07-03

**Soundness:** 4 excellent
**Presentation:** 3 good
**Contribution:** 4 excellent
**Rating:** 6
**Confidence:** 4

**Summary:**

This paper studies the matrix sensing problem in the semi-random model, where a subset of the matrices satisfy an RIP property and the rest are adversarially chosen so as to make the RIP not hold. The algorithm iteratively updates weights on the sensing matrices throughout the iterations to search for the set on which the RIP holds. Through this updating, the approximation error geometrically decreases.

**Strengths:**

- This work makes an important step towards provable methods when an RIP does not hold. It forms a natural and nontrivial extension of recent work on sparse recovery to the matrix sensing setting.
- The work introduces novel weighted and relaxed RIP conditions, under which it is still possible to efficiently identify and recover low-rank matrices.

**Weaknesses:**

- Only the noiseless setting is studied.
- The work is more expensive than nonconvex methods in the random model where the rank is known, which have complexity $O(ndr)$, whereas the proposed method has complexity $O(nd^{\omega}r)$.
- There are no experiments on data. Furthermore, it is unclear in what real/practical settings wRIP/dRIP might hold.

**Questions:**

- My main question is about how well the algorithm performs in practice. Can the authors give some experiments showing how well it actually runs?
- What challenges arise in the analysis that did not happen in the case of sparse vectors? What I'm asking is how does the analysis differ from the past work [KLL+22] to make it sufficiently novel/non-incremental, and not just a translation to the matrix recovery setting?
- The statement of algorithm 1 is not useful - there is only one step within the loop that calls another function, and everything else is just parameter initialization.
- The work might read better if vectors and matrices are distinguished from parameters somehow (for example, by using bold symbols).
- Can the authors give more intuition/explanation for the dRIP condition and how it is used? They mention that it would be explained more in Section 4 but I do not see it, other than it makes the oracle work.

**Limitations:**

- The work contains no experiments.
- The computational complexity of the method is not able to adapt to the rank of the underlying matrix $X^*$.
- The novelty of the theoretical methodology may be limited in how different it is from [KLL+ 22], but I have not checked that reference.

---

> ### Author Rebuttal · Authors · 2023-08-10
>
> We thank the reviewer for the thoughtful comments and positive feedback.
>
> On the challenges in the matrix case: While the generalization from vectors to matrices (by considering the singular values as a “vector”) may appear intuitive, such generalization should not be taken for granted. To illustrate, consider a simple equation $e^{a+b} = e^a e^b$, which continues to hold in the vector case $\sum_i e^{a_i + b_i} = \sum_i e^{a_i} e^{b_i}$, but its generalization to matrices is the well-known Golden–Thompson inequality $tr(e^{A+B}) \le tr(e^A e^B)$ which requires a non-trivial proof. We refer the readers to Section 2 of [ALO16], which pointed out several wrong matrix inequalities that found their way into published papers when people tried to generalize proofs to the matrix case. Concretely in our paper, we ran into some difficulties in proving our Lemma 5.3 (in Appendix B), and its proof required techniques that are distinct from the vector case. On another note, the algorithm in [KLL+23] runs in nearly-linear time, but despite our best efforts, we were not able to obtain the same runtime due to several technical obstacles.
>
> Regarding experiments: We believe that our results are substantial even without experiments, and we hope that our theoretical results are judged based on their merits. Our main contribution is to pose and study the problem of semi-random matrix sensing, and to provide descent-style algorithms with good asymptotic runtimes. We did not include experiments because our focus is not to obtain the fastest possible algorithm for semi-random matrix sensing, and experiments sometimes divert readers' attention toward such metrics. That said, we acknowledge that experimental evaluation is important and a fruitful direction for future work.
>
> On the remaining weaknesses and questions:
>
> - We chose to study the noiseless case because it is the most basic setting and our focus is the semi-random model (vs. the random setting with RIP). While our algorithm is simple, our analysis is already fairly involved, so we decided to leave the noisy setting to future work.
>
> - While it is true that our algorithm runs slower than the fastest non-robust matrix sensing algorithm (i.e., with RIP condition), our setting is inherently more challenging because we allow the adversary to add any number of arbitrary sensing matrices. Our goal is to initiate a line of research toward designing robust matrix sensing algorithms that are as efficient as their non-robust counterparts (or show that it is impossible to do so).
>
> - We believe Algorithm 1 helps improve the clarity of our paper. While Algorithm 1 appears trivial, it allows Algorithm 2 to focus on obtaining a matrix $X$ that is closer to the ground truth $X^\star$ by a constant factor. The alternative would be to put all its technical details into Algorithm 2 (including the for loop, the number of iterations, the contracting radius, and the union bound in failure probability), which would introduce unnecessary complexity in Algorithm 2.
>
> - Throughout the paper, we tried to use lower-case letters for vectors and upper-case letters for matrices. We notice that there are exceptions to this rule (e.g., L being a scalar) and we will try our best to address this.
>
> - Regarding more explanation on the dRIP condition and the weight oracle: In the proof of Lemma 4.2, we provided more intuition for the progress and decomposition conditions and explained how these conditions are used to prove the correctness of Algorithm 2. We apologize for any confusion caused by directing readers to Section 4, as we moved the proof of Lemma 4.2 to Appendix A in this submission. We will fix this. More details about the dRIP condition can be found in Appendix B.
>
> [ALO16] Zeyuan Allen-Zhu, Yin Tat Lee, Lorenzo Orecchia. Using Optimization to Obtain a Width-Independent, Parallel, Simpler, and Faster Positive SDP Solver. SODA 2016.
>
> [KLL+23] Jonathan A. Kelner, Jerry Li, Allen Liu, Aaron Sidford, Kevin Tian. Semi-Random Sparse Recovery in Nearly-Linear Time. COLT 2023.

---

### Official Review · Reviewer_ZDRS · 2023-07-10

**Soundness:** 3 good
**Presentation:** 3 good
**Contribution:** 3 good
**Rating:** 7
**Confidence:** 3

**Summary:**

The paper introduces a new descent algorithm for low rank matrix recovery in the setting of the semirandom model (where a subset of the measurements satisfy the RIP). The algorithm is based on a research of the best reweighting of the measurements (in order to navigate the non convex landscape) and a minimization of the data fidelity term.

**Strengths:**

There is clearly some novelty in the paper. Although as indicated it applies the ideas from Kelner et al 2022. It is sound and well written.

**Weaknesses:**

General comments

It is a good extension of the idea in Kelner et al 2022. The introduction of the algorithm could be improved though. The algorithm is clearly and neatly stated but a clear intuition on the main steps would be a clear addition to the paper. E.x. the relation between the weight oracle and lemma 4.2 is not completely clear. The weight oracle is key and some additional intuition on why it enables a contraction of the distance to the ground truth would be helpful (see detailed comments below). I..e as you indicate, it is hard to reverse the action of the adversary so what makes the algorithm efficient ? I.e. it seems to me you are leveraging the diversity of the measurements to escape local minimas. Why does the weighted objective enable you to escape the local mins introduced by the non RIP part of the matrix? your gradient is a simple LS gradient so the weights are really key. This is my main criticism with the paper. See my comments below on the progress and decomposition guarantees. If I understand well, the weight oracle gives you a sufficiently large “escape” direction. Yet this only does not imply convergence. convergence comes from key lemma 4.2 which if I understand well implies that the existence of such a sufficiently large escape direction will lead to a gradient step that will decrease the distance to the global solution. Unfortunately there is almost no intuition provided for this lemma.

Finally, I think you could better summarize your contribution with one sentence or two of the form “The algorithm relies on a combined optimisation of the weight (in order to escape non convexity) and minimization of the cost function (in order to get closer to the true minimizer)”
You don’t have to explicitly use the above but some clear intuition on why the algorithm is working is definitely missing. In particular, you need to provide (some more) intuition on lemma 4.2.


Detailed comments

page 2
 - line 33-38, you should directly detail how you generate the non RIP matrices. Also, if your measurements are generated as the union of the RIP and non RIP matrices, then I don’t see why having additional non RIP measurements will be a problem. You just add redundant constraints. If you use a subset of the union then you should say it clearly
- Is finding a submatrix that satisfies RIP is hard ? if so you should say it as well because otherwise, you could just retrieve the RIP part of the matrix and solve the problem from there.

page 3
- You should improve the statement of Theorem 1.1. recall the meaning of d, n and omega. Do you have convergence regardless of the number of adversarial measurements? this should also be clarified in the statement
- Ok so as I indicated above, the first paragraph in section 1.2 (where you explain that it is NP hard to extract a RIP matrix) should appear earlier (or at last you should mention the hardness of the RIP submatrix detection problem in one sentence in the abstract or/and introduction)
- lines 112 -122, you mention the “progress” and “decomposition” gaurantees. It would help to have a decomposition of the paragraph with two clear items providing a short explanation on each guarantee

page 5
- When you introduce the distinction between the RIP and dRIP conditions. It is not clear why reweighting the matrices will be useful. Perhaps you could add a sentence explaining why the reweighting is important/how it will affect recovery.
- Generally speaking it is good to indicate that the isometry condition in the wRIP is a relaxation of the isometry condition in the RIP, it could be good to indicate that the wRIP can be understood as requiring that only a subset of the measurements satisfy the RIP

page 6
- The halveError subroutine in algorithm 1 should be introduced earlier. In fact I’m not sure Algorihtm 1 is essential here
- I know you mention it on line 184 but the sample complexity is important for the matrix sensing and completion problem and they should appear in the statement of Theorem 3.1.
- It is a detail byut the oracle which is named Oracle in Algorithm 3 is named O in Algorithm 2
- When you introduce the Progress and decomposition guarantees which are key to the algorithm, you indicate that “the purpose of those” will be further discussed in section 4. Yet section 4 barely adds to the definition of the properties, merely indicating that success of the oracle corresponds to satisfaction of the progress guarantee which in turns leads to a reduction of the distance to the groundtruth . We would like more intuition on why the weights make such a difference in the recovery.
- Generally speaking when you introduce the “progress” and “”decomposition” guarantees, those are equivalent to finding a weight so that there is a large gradient.

page 8

- lines 303 - 306 “which means with high probability the weight oracle produces output satisfying the progress and decomposition guarantees then each iteration decreases the distance” —> Why is that clear ?


typos:
- page 3: lines 108-109, “Ideally, the property should” —> “Ideally the properties should be (1) … (2)…”
- page 3 line 109 “ensure the weighted gradient step makes ” —> “ensure that the gradient step …” would be more clear
- same page, line 109 again: “secondly, can …” —> “secondly ensure that those steps can …”
- page 8, line 304 “the weight oracle produces output ..” —> “the weight oracle produces an output ..”



**Questions:**

see above

---

> ### Author Rebuttal · Authors · 2023-08-10
>
> We thank the reviewer for the thoughtful comments and positive feedback.
>
> Regarding how the non-RIP matrices are generated and why they may cause issues: The non-RIP matrices can be arbitrary. In fact, a computationally-unbounded adversary can generate (any number of) additional sensing matrices $A_i$ after examining the ground-truth matrix, the good (RIP) sensing matrices, and our algorithm. The only constraint is that the corresponding linear measurements $b_i = \langle A_i, X^\star \rangle$ must be accurate where $X^\top$ is the ground-truth matrix to be recovered. These additional sensing matrices indeed only add redundant constraints for convex approaches (e.g., nuclear norm minimization), but they pose a series risk for non-convex approaches, because they can destroy the *landscape* of many non-convex objectives. More specifically, all local optima are globally optimal with RIP, but now these extra sensing matrices may introduce bad local optima.
>
> Regarding the purpose of the progress and decomposition guarantees (the reviewer’s 4th comments for Page 6): In the proof of Lemma 4.2, we provided more intuition for the progress and decomposition conditions and their role in proving the correctness of Algorithm 2. We apologize for any confusion caused by directing readers to Section 4, as we moved the proof of Lemma 4.2 to Appendix A in this submission. We will fix this.
>
> On the correctness of Lemma 4.2 (the reviewer’s comments for Page 8): This follows from Lemma 5.3. We will make this clear.
>
> We appreciate the reviewer providing many detailed suggestions on how we could improve the presentation of our paper:
>
> - Define $d$, $n$, and $\omega$ in Theorem 1.1, and highlight that the adversary can add any number of arbitrary sensing matrices.
>
> - Add an earlier statement on checking the RIP condition is NP-Hard.
>
> - Provide more intuition on the progress and decomposition guarantees and how they affect the recovery of the ground-truth matrix.
>
> - Emphasize that the wRIP condition is equivalent to requiring only a subset of the sensing matrices satisfying RIP.
>
> - Discuss the sample complexity (i.e., the number of random matrices needed to satisfy RIP with high probability) in or around Theorem 3.1.
>
> - Name the weight oracle consistently in Algorithms 2 and 3.
>
> We will address all of them.
>
> We thank the reviewer for pointing out the typos in our paper. We will fix them.

---

### Decision · Program_Chairs · 2023-09-21

**Decision:**

Accept (poster)

**Comment:**

This paper studies the matrix sensing problem with a semi-random measurement model, where a subset of the measurement matrices are norm-preserving (i.e., they satisfy the RIP condition), while the remaining measurement matrices are chosen adversarially. With the addition of adversarially chosen measurement matrices, the combined measurement operator may not satisfy RIP condition. This implies that the existing results on the benign landscape of matrix sensing do not apply to this setting. To address this challenge, the authors introduce a weighted gradient-based method that can guarantee convergence to the true low-rank solution in polynomial time.

Four reviewers have reviewed the paper, and their overall assessment of the paper was positive. I agree with this assessment and believe the paper is a solid contribution with interesting results. Three reviewers also recommended that the authors should provide some experiments to showcase the empirical performance of their proposed algorithm. I agree with this comment, and encourage the authors to include (at least some preliminary) experiments in the final version of the paper.